# Description and Evaluation of the Specified-Dynamics Experiment in the Chemistry-Climate Model Initiative (CCMI)

Clara Orbe[1], David A. Plummer[2], Darryn W. Waugh[3], Huang Yang[3,4], Patrick Jöckel[5], Douglas E. Kinnison[6], Beatrice Josse[7], Virginie Marecal[7], Makoto Deushi[8], Nathan Luke Abraham[9,10], Alexander T. Archibald[9,10], Martyn P. Chipperfield[11], Sandip Dhomse[11], Wuhu Feng[10,11], and Slimane Bekki[12]

[1]NASA Goddard Institute for Space Studies, New York, NY, USA
[2]Climate Research Branch, Environment and Climate Change Canada, Montreal, QC, Canada
[3]Department of Earth and Planetary Sciences, Johns Hopkins University, Baltimore, Maryland, USA
[4]Department of Atmospheric and Ocean Sciences, University of California, Los Angeles
[5]Deutsches Zentrum für Luft- und Raumfahrt (DLR), Institut für Physik der Atmosphäre, Oberpfaffenhofen, Germany
[6]National Center for Atmospheric Research (NCAR), Atmospheric Chemistry Observations and Modeling (ACOM) Laboratory, Boulder, USA
[7]Centre National de Recherches Météorologiques UMR 3589, Météo-France/CNRS, Toulouse, France
[8]Meteorological Research Institute (MRI), Tsukuba, Japan
[9]Department of Chemistry, University of Cambridge, Cambridge, CB2 1EW, UK
[10]National Centre for Atmospheric Science, UK
[11]School of Earth and Environment, University of Leeds, Leeds, UK
[12]Laboratoire de Météorologie Dynamique (LMD/IPSL)

*Correspondence to:* Clara Orbe (clara.orbe@nasa.gov)

**Abstract.**

We provide an overview of the REF-C1SD "Specified-Dynamics" experiment that was conducted as part of Phase 1 of the Chemistry-Climate Model Initiative (CCMI). The REF-C1SD experiment, which consisted of mainly nudged general circulation models (GCMs) constrained with (re)analysis fields, was designed to examine the influence of the large-scale circulation on past trends in atmospheric composition. The REF-C1SD simulations were produced across various model frameworks and are evaluated in terms of how well they represent different measures of the dynamical and transport circulations. In the troposphere there are large ($\sim 40\%$) differences in the climatological mean distributions, seasonal cycle amplitude and trends of the meridional and vertical winds. In the stratosphere there are similarly large ($\sim 50\%$) differences in the magnitude, trends and seasonal cycle amplitude of the Transformed Eulerian Mean circulation and among various chemical and idealized tracers. At the same time, interannual variations in nearly all quantities are very well represented, compared to the underlying reanalyses. We show that the differences in magnitude, trends and seasonal cycle are not related to the use of different reanalysis products; rather, we show they are associated with how the simulations were implemented, by which we refer both to how the large-scale flow was prescribed and to biases in the underlying free-running models. In most cases these differences are shown to be as large or even larger than the differences exhibited by free-running simulations produced *using the exact same models*, which are also shown to be more dynamically consistent. Overall, our results suggest that care must be taken when using specified-dynamics simulations to examine the influence of large-scale dynamics on composition.

# 1 Introduction

Understanding the interaction between large-scale dynamics and atmospheric composition is important for understanding the past and future behavior of greenhouse gases (GHGs) and ozone-depleting substances (ODS). However, biases in large-scale atmospheric transport (both in terms of climatological means and interannual variability) remain large sources of uncertainty when assessing simulations of atmospheric composition. One approach to reduce this uncertainty has been to use numerical models that are constrained with meteorological fields taken from (re)analysis products. In this spirit, chemistry climate models (CCM) participating in the Chemistry-Climate Model Initiative (CCMI, Eyring et al. (2013)), were asked to perform a so-called "Specified-Dynamics" simulation of the recent past (1980-2009) as part of the CCMI Phase1 Hindcast experiment using large-scale flow fields taken from meteorological analyses and observed sea surface temperatures (SSTs) and sea ice concentrations (SICs). Modeling groups also performed parallel free-running integrations of the recent past using the same models and boundary conditions (i.e. SSTs and SICs).

While specified-dynamics simulations are commonly used in studies of atmospheric composition, it is not obvious that using analyzed meteorological fields necessarily improves simulation of the transport circulation (i.e. the "tracer-independent" properties of the flow (Holzer and Hall (2000)) that are often summarized using measures like the stratospheric mean age (Hall and Plumb (1994), among others). This is not only because of differences among reanalysis products, which can be large, especially in the stratosphere (e.g. Seviour et al. (2011), Abalos et al. (2015)), but also because of the various ways in which a model may be constrained to analysis fields. Studies have long shown that transport computations using analyzed winds are very sensitive to how the large-scale flow is specified (e.g. Schoeberl et al. (2003), Meijer et al. (2004), Pawson et al. (2007)). However, for historical reasons these sensitivities have been most rigorously explored in the context of offline chemical transport models (CTMs) and Lagrangian trajectory models, with several studies demonstrating the sensitivity of stratospheric transport to both the temporal sampling and averaging of the prescribed fields (e.g., Waugh et al. (1997), Bregman et al. (2006), Legras et al. (2004), Pawson et al. (2007), Monge-Sanz et al. (2007, 2012, 2013)). By comparison, relatively less attention has been paid to assessing the credibility of large-scale transport in simulations using general circulation models constrained with reanalysis products using either so-called "nudging," wherein the simulated meteorological fields are relaxed towards analysis fields (Kunz et al. (2012)), or using approaches derived from data assimilation (e.g. Orbe et al. (2017b)). While these studies have demonstrated large sensitivities in simulated transport to (at times arbitrary) choices in how the nudging is applied (e.g. Orbe et al. (2017b)) it is difficult to draw general conclusions as it is not clear how specific these findings are to the particular model used and/or nudging approach implemented.

In addition to the lack of studies focused on evaluating simulated transport in nudged simulations, most intercomparisons focusing on atmospheric composition have primarily utilized CTMs (e.g. the Task Force on Hemispheric Transport of Air Pollution (HTAP) (HTAP, UNECE LTRAP (2007)) and the Atmospheric Tracer Transport Model Intercomparison (TransCOM) (Patra et al. (2011)) or free-running simulations (e.g. the Atmospheric Chemistry and Climate Model Intercomparison Project (ACCMIP) (Lamarque et al. (2013)) and the SPARC Chemistry-Climate Model Validation (CCMVal) (Eyring et al. (2008)). Thus, to the best of our knowledge, no intermodel comparison prior to CCMI has provided the output and experiments needed

to rigorously evaluate the representation of dynamics and transport in primarily online "nudged" simulations. (Note that by "online" simulations we refer to those that have been produced using general circulation models.

Another novelty of CCMI is that modeling centers provided both hindcast specified-dynamics and free-running simulations – herein referred to as REF-C1SD (simply SD for "specified dynamics") and REF-C1 (simply FR for "free-running"), respectively – which presents a unique opportunity to compare the performance of specified-dynamics simulations relative to free-running integrations produced *using the exact same versions of the models*. Indeed, recent inquiries in this vein have proved illuminating, with Orbe et al. (2018) showing that the differences in interhemispheric transport (IHT) among the SD simulations are as large as the differences among FR integrations produced using the same models. More recently, Yang et al. (2019) analyzed the differences among tracers with more realistic anthropogenic emissions than those considered by Orbe et al. (2018), who focused only on tracers with zonally uniform sources, and also showed large differences in transport among the SD simulations. Unlike in Orbe et al. (2018), who focused primarily on IHT differences in the context of parameterized convection in the tropics, Yang et al. (2019) focused on transport from NH midlatitudes into the Arctic. Furthermore, they associated the spread in transport among the SD simulations to differences in the large-scale flow, specifically the poleward extent of the Hadley Cell, evaluated in that study in terms of the near surface meridional wind. This finding is particularly surprising, given that the meridional winds were specified in these simulations, albeit using a broad range of nudging techniques and sources of meteorological fields.

The findings presented in Orbe et al. (2018) and Yang et al. (2019) provide only a limited comparison of the large-scale flow fields among the REF-C1SD ensemble. More importantly, they provided no details about how the REF-C1SD simulations were actually implemented among the different models groups, information that is difficult – if not impossible – to access in the published literature. The goals of this study, therefore, are two-fold: 1) document how the specified-dynamics hindcast simulations were implemented and 2) quantify key differences in first-order measures of the tropospheric and stratospheric dynamical and transport circulations. Via 2) our goal is to present a more comprehensive evaluation of the large-scale flow than presented in Orbe et al. (2018) and Yang et al. (2019) and to extend our analysis to the stratosphere, which has been evaluated in CCMI models primarily using the free-running REF-C1 experiment (Dietmuller et al. (2018)). Note that, while Chrysanthou et al. (2019) presented the first comparison of the stratospheric residual circulation among the nudged CCMI Hindcast runs, our analysis, which complements the findings presented in that study, has a broader scope by focusing on both the troposphere and the stratosphere and including discussions of large-scale transport.

It is important to note at the outset that there are several potential sources of differences among the SD simulations: 1) the use of different reanalysis fields, 2) differences in how the large-scale flow is constrained and 3) differences associated with biases in the underlying free-running models used to produce the SD simulations. When possible we try to isolate which source is most likely responsible for the spread among the simulations but since 2) and 3) are in practice often related, and thus difficult to isolate from each other, we refer to them both using the general phrase "implementation differences." We begin by discussing the models used and output analyzed in Section 2 and various aspects about how the simulations were implemented in Section 4, followed by a comparison of key large-scale dynamical and transport properties in Sections 4 and 5. Brief conclusions in Section 6 are followed by details specific to each individual model (Appendix).

## 2 Methodology

### 2.1 Models and Experiments

The CCMI Hindcast experiment consisted of both free-running REF-C1 (FR) and specified-dynamics REF-C1SD (SD) simulations, both of which were constrained with observed SSTs and SICs. Here we report the details of how the SD simulations were implemented among models, based on feedback we received in response to a survey that was distributed among CCMI model contact leads. Among those simulations we only show results from output that was uploaded to the British Atmospheric Data Centre (BADC) archive (ftp://ftp.ceda.ac.uk) and/or provided to us via personal communication. Output from the WACCM and CAM simulations was obtained from the NCAR Earth System Grid portal (https://www.earthsystemgrid.org/). Table 1 lists the modeling groups that responded to our survey, what type of model was used for the SD simulation (offline CTM or online nudged CCM), and which source of meteorological fields was used. We also note whether a parallel free-running simulation was performed, since the subset of models for which both FR and SD simulations were performed comprises a unique ensemble within the Hindcast experiment (hereafter referred to with an asterisk as SD*) that is ideal for evaluating the performance of specified-dynamics simulations relative to free-running simulations.

We note that typically only one SD experiment was submitted per modeling group. However, as in Orbe et al. (2017a, 2018) and Yang et al. (2019) we include two SD simulations from NASA and NCAR, denoted in all figures using a color convention that is similar to what was used in those studies. In particular, the two NASA SD simulations here refer to an offline integration of the NASA Global Chemical Modeling Initiative (GMI) Chemical Transport Model (Strahan et al. (2007, 2016)) as well as an online simulation of the Global Earth Observing System (GEOS) general circulation model, both constrained with MERRA meteorological fields (Reinecker et al. (2011)) (not MERRA-2 (Gelaro et al. (2017))). We also present two SD NCAR simulations in which WACCM was nudged to MERRA on two different relaxation timescales. Further details of those (and all other) simulations are presented in the Appendix. Finally, in addition to differences among the REF-C1 and REF-C1SD experiments, the models differ widely in terms of their horizontal resolution, vertical resolution, and choices of sub-grid scale (i.e. turbulence and convective) parameterizations. For a more comprehensive review of these details we refer the reader to Morgenstern et al. (2017). In all cases only a single simulation was taken from the REF-C1 experiments for models that submitted multiple ensemble members. This was usually the "r1i1p1" simulation; the only exception to this was the CNRM-CM 5-3 simulation, for which only the "r1i1p2" output was available on the BADC archive.

### 2.2 Diagnostic Output

While the primary purpose of this study is to document how the REF-C1SD experiment was implemented among models (Section 3) we also take the opportunity to provide more extensive comparisons of the large-scale flow and transport fields than what was shown by Orbe et al. (2018) and Yang et al. (2019) (Table 2). In order to compare the flow among the models we focus on basic first-order measures, including the three-dimensional winds (U, V and $\omega$) and temperatures (T) in the troposphere. (Note that the vertical velocity (w) was available in all simulations and then converted into pressure velocity through the relation $\omega = -\mathrm{w}\rho\mathrm{g}$, where $\rho$ and $g$ are density and gravity, respectively.) In the stratosphere, the dynamical circulation is more naturally

quantified using the Transformed Eulerian Mean (TEM) residual meridional ($v^*$) and vertical ($w^*$) velocities (Andrews et al., 1987). Following Dietmuller et al. (2018) we note that since $w^*$ was calculated slightly differently among models, specifically with respect to the conversion of the Lagrangian tendency of air pressure, we also derived $w^*$ independently from $v^*$ by continuity, as in that study. Comparisons of $w^*$ between the model output and the values inferred from $v^*$ are presented in Section IV and result in no major differences with respect to our main findings.

In addition to circulation diagnostics we also include comparisons of ozone ($O_3$), nitrous oxide ($N_2O$) and the stratospheric mean age ($\Gamma_{STRAT}$)– the mean transit or "elapsed" time since air last contacted the tropical tropopause (i.e. Hall and Plumb (1994), Waugh and Hall (2002)). We also present comparisons in the stratosphere of the Northern Hemisphere (NH) midlatitude mean age ($\Gamma_{NH}$), defined as the mean transit time since air last contacted the NH midlatitude surface (Waugh et al. (2013), Orbe et al. (2018)), since few models integrated both the stratospheric and NH midlatitude mean age tracers (Table 2). Thus, while they physically capture similar aspects of the transport circulation, the output from the different age tracers comprise different groups of models within the larger SD ensemble and therefore provide relatively independent perspectives on the transport differences among the simulations. We make the ages more comparable by subtracting off a reference mean age value, evaluated here as the mean age at 100 hPa, averaged over 10°S to 10°N. This also corrects for the fact that the stratospheric mean age tracer was implemented differently among different models, with some models applying the lower boundary condition globally at the surface (versus only in the tropics). A similar approach was used by Dietmuller et al. (2018), except their reference was defined relative to the tropical tropopause in each model.

For our analysis of tropospheric variables we interpolated all output from native model levels to a standard pressure vector with four pressure levels in the stratosphere (10, 30, 50 and 80 hPa) and 19 pressure levels in the troposphere spaced every 50 hPa between 100 hPa and 1000 hPa (Orbe et al. (2018)). Unlike in the troposphere, the stratospheric circulation and tracer output was requested on 31 constant pressure surfaces from 1000 to 0.1 hPa so no additional interpolation in the vertical was required. However, for some models (i.e. MOCAGE, CAM, WACCM, MRI) the output was available on different pressure levels that had to be interpolated manually to the 31 pressure levels. For both tropospheric and stratospheric variables we also interpolated in the horizontal to the same 1° latitude by 1° longitude grid as in Orbe et al. (2018). Only monthly mean output is used as that is all that was available for the quantities analyzed here.

Finally, when possible we compare the output from SD simulations with fields from ERA-Interim (hereafter ERA-I) (Dee et al. (2011)), MERRA (Reinecker et al. (2011)), and JRA-55 (Kobayashi et al. (2015)). For the case of the TEM circulation components, $v^*$ and $w^*$, we have used the common grid (2.5° latitude by 2.5° longitude) output from the SPARC Reanalysis Intercomparison Project (S-RIP) dataset (Martineau et al. (2018)). Note that only $\omega^*$ was available from the S-RIP fields (units: Pa/s), whereas the CCMI output is in terms of $w^*$ (units: m/s), thus requiring that we convert the S-RIP fields to $w^*$ using the following relation: $\overline{w}^* = -\frac{H}{p}\overline{\omega}^*$, where H = $RT_s$/g is a mean scale height of the atmosphere, here taken to be 7 km, corresponding to $T_s \sim$ 240K, a constant reference air temperature (see equation A16 in Gerber and Manzini (2016)).

## 2.3 Metrics

For all variables we first compare ten-year 2000-2009 climatological mean meridional profiles among the SD simulations and among the reanalysis fields (when available) to which they were initially constrained (Section 4.1). Then we compare the temporal variability of the simulations, first comparing the seasonal cycle amplitude (SCA) and phase among the simulations, also for the 2000-2009 period, both with respect to the other simulations and with respect to the reanalysis products (Section 4.2). As in Barnes et al. (2016) we define the SCA as the climatological seasonal cycle of the zonally-averaged fields at every pressure level and latitude (note we do not apply their 31-day filter as they were using daily data and we are using monthly data). The SCA is then defined as the difference between the maximum and minimum of the seasonal cycle, respectively designated throughout as $\tau_{max}$ and $\tau_{min}$. In addition, throughout we normalize the SCA by its climatological annual mean value in order to account for the fact that for some variables the seasonal cycle is small so that discrepancies in (unnormalized) SCA values may appear larger than they actually are, relative to the climatology. Care is taken throughout to identify those cases when the seasonal cycle amplitude is small.

In addition to the seasonal cycle we also assess how well the simulations covary with each other on interannual timescales over years 1980-2009. As such, our assessment of interannual variability, which evaluates only the degree of correlation between timeseries, differs from previous studies (Chrysanthou et al. (2019)), in which timeseries were further decomposed in terms of different modes of interannual variability (i.e. the El-Nino Southern Oscillation, the Quasi-Biennial Oscillation, etc.). More precisely, for each member within the SD ensemble we identify a given variable $\chi$ for which we first remove the linear trend and then calculate the correlation coefficient between the annual mean time series corresponding to that ensemble member $i$ and the annual mean time series of its corresponding ensemble mean. For example, $\mathrm{corr(i)}_{\mathrm{U,ERA}}$ corresponds to the correlation coefficient between the detrended annually averaged zonal winds of simulation $i$ and the (also detrended and annually averaged) zonal winds averaged over the ensemble of simulations constrained with ERA-Interim fields. We also evaluate how well each simulation varies relative to the entire SD ensemble, the mean of which will average out differences among the reanalysis products, denoted hereafter as $\mathrm{corr(i)}_{\chi,\mathrm{SD}}$. Correspondingly, for any given ensemble of SD simulations $M$ (e.g. ERA-I, MERRA, JRA-55, SD) consisting of $N$ members the ensemble mean of $\mathrm{corr(i)}_{\chi,\mathrm{M}}$ is throughout denoted as $\overline{\mathrm{corr}}_{\chi,\mathrm{M}} \equiv \frac{1}{N}\sum_{i=1}^{N}\mathrm{corr(i)}_{\chi,\mathrm{M}}$.

Finally, we conclude our analysis by comparing trends of the various measures, evaluated here simply as the linear fit to the deseasonalized time series. While previous studies have already cautioned about the suitability of inferring trends from either reanalyses (Abalos et al. (2015)) or nudged experiments (Chrysanthou et al. (2019)) those studies focused on the lower stratosphere (specifically, the Transformed Eulerian mean circulation) and it is not clear to what extent their conclusions about trends apply to other dynamical and constituent fields.

## 3 Implementation of the REF-C1SD Simulation

Here we summarize key aspects describing how the SD simulations were implemented. For more detailed descriptions of implementation in the individual models we refer the reader to the Appendix. As such, both sections complement the information provided in Table S30 in the supplementary material of Morgenstern et al. (2017).

### 3.1 Nudging versus CTM

Most of the REF-C1SD simulations were performed as nudged simulations using online CCMs (MRI-ESM1r1, GEOS, HadGEM3-ES, GFDL-AM3, EMAC, CNRM-CM 5-3, CHASER (MIROC-ESM), CESM WACCM, CESM1 CAM4-chem, CCSRNIES MIROC3.2, IPSL, UMUKCA, CMAM) although a few groups also submitted results from offline CTMs (TOMCAT CTM, MOCAGE CTM, and NASA GMI-CTM). Note that the GEOS REF-C1SD simulation did not use a standard nudging approach but, rather, the "replay" approach, which involves reading in MERRA fields and recomputing the analysis increments, which are applied as a forcing to the meteorology at every model time step (Orbe et al. (2017b) (see Appendix for more information)). In addition, note also that output from the HadGEM3-ES, and GFDL-AM3 simulations was not available so our analysis comprises a total of 13 simulations produced using online models and 3 simulations produced using CTMs.

### 3.2 Sources of Meteorological Fields

Large-scale meteorological fields from the three reanalysis products ERA-I, MERRA and JRA-55 were used to constrain the REF-C1SD simulations. Although NCEP/NCAR (Kistler et al. (2001)) fields were also used in the GFDL simulations those fields are not analyzed here since that output was not available. Among the ERA-I constrained simulations, all use six-hourly instantaneous fields, although differences may still arise among those simulations due to differences in how the analysis fields were interpolated to the models' native grids. By comparison, among the MERRA-constrained simulations, there are additional differences related to the fact that multiple MERRA products were used. In particular, while the GMI CTM simulation used the three-hourly time-averaged assimilated fields, the GEOS and WACCM REF-C1SD simulations were constrained using six-hourly instantaneous analysis fields. An examination of the differences in stratospheric transport implied by using assimilated versus analysis fields for the GEOS model was presented in Orbe et al. (2017b). Specifically, within the context of "replay" they showed that the use of analysis fields produced stratospheric mean age values that were consistently younger, than if assimilated fields were used, irrespective of their temporal sampling (three-hourly vs. six-hourly).

### 3.3 Boundary Conditions

Most of the REF-C1SD simulations use the Hadley Centre Ice and Sea Surface Temperature (HADISST) dataset (Rayner et al. (2003)), as recommended. Some simulations, however, were forced with climatological SSTs and SICs taken from other datasets derived both from models (TOMCAT CTM) and other observational sets. The latter include simulations constrained with monthly mean SSTs as used in the AMIP simulations and described in Hurrell et al. (2008) (IPSL LMDz-REPROBUS).

The former include simulations that were constrained with SSTs from Reynolds et al. (2000) (the NASA GMI-CTM simulation) and ERA-Interim (EMAC).

### 3.4 Constrained Variables, Nudging Spatial Domains and Relaxation Timescales

Two major sources of differences among the nudged simulations are the choice of large-scale fields and nudging timescales with which the model fields were constrained to the analyses. For example, while nearly all the REF-C1SD simulations are constrained to the east-west and north-south components of the horizontal wind (U,V) some models (EMAC, TOMCAT) were nudged to the divergence and vorticity fields. In addition, several simulations also nudged to temperature (T) (or potential temperature, as in UMUKCA), water vapor, (the logarithm of) surface pressure, surface stress, and latent and sensible heat fluxes. A few models also applied nudging in spectral space (EMAC).

Nudging timescales and nudging domains varied widely among the different simulations, where we define the nudging relaxation time constant $\tau$ such that the nudging increment for variable $\chi$ is proportional to $(\chi_{analysis} - \chi_{model})/\tau$ (note that $\tau$ has units of hours). In particular, $\tau$ ranged from as low as 5 hours in some simulations (CNRM-CM 5-3, WACCM 5-hr) to as long as 60 hours in others (GFDL-AM3), with some simulations applying spatially uniform nudging (e.g. CMAM and UMUKCA) while in others $\tau$ depends explicitly on pressure or model level (GFDL-AM3, MRI-ESM1r1, EMAC).

### 3.5 Sources of Convective Mass Fluxes

In addition to differences in the resolved flow among the simulations, another large source of differences are the (parameterized) convective mass fluxes used to simulate convective transport. These were either taken from the same analysis dataset from which the large-scale flow fields were obtained (the NASA GMI-CTM) or recalculated online using the model's own convective parameterization. The latter approach was used mainly in the nudged simulations, although some offline models also recomputed the convective mass fluxes.

A broad range of convective parameterizations are used including relaxed and/or triggered schemes as described in Moorthi and Suarez (1992) and Zhang and McFarlane (1995) (CESM1 CAM4-chem and WACCM, CMAM, GMI-CTM), nearly instantaneous adjustment schemes along the lines of Arakawa and Schubert (1974) (CCSRNIES MIROC3.2, CHASER MIROC-ESM, MRI-ESM1r1), as well as diagnostic closure schemes based on large-scale moisture or mass convergence similar to Tiedtke (1989) (TOMCAT CTM, EMAC).

## 4    Comparisons of Large-Scale Flow Among REF-C1SD Simulations

We now present a comparison of various large-scale flow measures of the REF-C1SD simulations, both in terms of their climatological mean distributions as well as their seasonal and interannual variability. Throughout special focus will be placed on interpreting the sources of differences among the REF-C1SD simulations, specifically with respect to the use of different reanalysis fields versus differences related both to how the large-scale flow fields are constrained as well as to underlying

free-running model biases. To this end, Table 3 summarizes these three potential sources of spread among the SD ensemble and highlights key examples of each source as demonstrated by the simulations analyzed in this study.

## 4.1 Climatological Distributions

We begin by comparing meridional profiles of the 2000-2009 climatological mean zonally averaged zonal winds at 850 hPa and 300 hPa, respectively chosen in order to evaluate the representation of the near-surface eddy-driven component of the zonal winds over midlatitudes and the subtropical jet (e.g. Barnes and Polvani (2013)). As shown in Figure 1 $U_{850}$ and $U_{300}$ compare very well among the SD simulations (Fig. 1a). Comparisons of the temperature field also reveal only small ($\sim$ 1-2 K) differences among the SD simulations (Fig. 1b), with the exception of one outlier (i.e. IPSL SD). Further inspection of that simulation confirms that it was nudged to ERA-I U, V and T using a height-dependent nudging timescale, with weaker nudging at lower pressures. This may explain why the biases in that outlier are larger in the upper troposphere, with values of $T_{300}$ in that simulation corresponding well with values from the free-running simulation produced using that same model (Supplementary Figure 1, Table 3, rows 2 and 3). That outlier aside, overall we conclude that the climatological zonal mean distributions of both the zonal winds and temperature fields are well constrained in the troposphere in the SD simulations, relative both to the reanalysis products and to the others members within the SD ensemble.

By comparison to the zonal winds and temperatures, the meridional winds (Fig 1c) reveal substantially larger differences among the SD simulations, with differences in $V_{850}$ approaching 0.4 m/s in the tropics and Southern Hemisphere (SH) midlatitudes and almost 1 m/s over NH midlatitudes (Fig. 1c, bottom) (Yang et al. (2019)). In the upper troposphere the differences in $V_{300}$ are equally as large, peaking at 0.4 m/s (or nearly 80% of the ensemble mean climatology) in the tropics and 0.3 m/s (also 80% of the climatological ensemble mean value) over the NH subtropical jet. Furthermore, although there are large differences among the reanalysis products, especially between MERRA versus ERA-I and JRA-55 in the tropics (Table 3, row 1), the differences among the SD simulations cannot be entirely understood in terms of the different reanalyses. Rather, a large fraction of the SD ensemble spread in V is spanned solely by simulations constrained with ERA-I fields (note that the differences among the MERRA and JRA-55 simulations are also large but appear smaller partly because those subsets of SD simulations contain fewer members).

In addition to the meridional winds we also find large differences in $\omega_{850}$ approaching 0.02 hPa/s in the tropics and 0.01 hPa/s in the subtropics, or $\sim$60% and $\sim$50% relative to the ensemble mean climatologies, respectively (Fig 1d, bottom). The differences aloft captured by $\omega_{300}$ are similar in magnitude (Fig 1d, top). As with the meridional winds the largest differences occur in the (sub)tropics and are not obviously related to differences associated with the use of different reanalysis products, although we note that $\omega$ was not part of the MERRA assimilated (ASM) collection analyzed here, which limits our interpretation somewhat. Furthermore, in all simulations $\omega$ was computed online and was not constrained directly to the reanalysis fields. Therefore, unlike the fields U, V and T, the differences in $\omega$ among the SD simulations not only reflect differences associated with the source of analysis fields but also the way in which $\omega$ was calculated online in models.

Comparisons of meridional profiles of the TEM circulation and the chemical and passive tracer distributions in the stratosphere (Figure 2) reveal that, overall, the differences in the TEM circulation among the SD simulations are even larger than the

meridional and vertical wind differences in the troposphere. Specifically, values of $v^*_{30\text{hPa}}$ range between -0.1 and 0.1 m/s in the subtropics (Fig. 2a, top) while differences in $v^*_{80\text{hPa}}$ approach 0.4 m/s over northern and southern midlatitudes (or $\sim 100\%$ the climatological mean ensemble mean value) (Fig. 2a, bottom). Similarly, the differences in $w^*_{80\text{hPa}}$ (Fig. 2b) approach $\sim 0.0008$ m/s (also $\sim 100\%$ the ensemble mean climatological value). Chrysanthou et al. (2019) noted similarly large differences in $w^*$ among the REF-C1SD simulations, although they examined a slightly different region in the stratosphere (10 hPa and 70 hPa versus the 30 hPa and 80 hPa pressure levels examined here).

As described earlier, the differences in the $w^*$ fields may be exaggerated by the fact they also potentially reflects inconsistencies in how that calculation was performed among modeling groups. Therefore, we also derive $w^*$ from continuity as outlined Dietmuller et al. (2018) and, consistent with their results, we find that the independent derivation of $w^*$ using $v^*$ does produce noticeable, and even larger, differences in the values of $w^*$ (Supplementary Figure 2). (Note that, in order to facilitate comparisons with Dietmuller et al. (2018) (specifically their Supplementary Figure S2) we also show averages over 20°S and 20°N). However, although we find that the absolute values of $w^*$ differ between the output provided on the BADC archive versus our offline calculations inferred from $v^*$, we nonetheless find that the *differences* in $w^*$ are of similar magnitude across the SD ensemble, irrespective of which calculation is used. Therefore, despite potential inconsistencies in how $w^*$ was calculated among modeling centers, the fact that it differs widely among SD simulations is a robust result.

Although for some variables and locations (e.g. $w^*$ at 30 hPa (Fig. 2b, top)) the TEM circulation values are clustered by reanalysis product, thus indicating that differences in the simulations appear to be primarily driven by differences among the reanalyses, this does not generally hold across variables and different locations in the stratosphere. This is particularly true for the chemical and idealized tracers, including $O_3$ at 80 hPa (Fig. 2c, bottom), with values spanning nearly 100 ppb (or 30% the climatological mean value), and for $N_2O$ at 30 hPa (Fig. 2d, top), for which differences over southern midlatitudes approach $\sim 100\%$ the climatological mean value (Table 3, row 3). For both tracers the ensemble spread is spanned nearly entirely by the ERA-I ensemble, although, as discussed earlier, this may simply reflect the overrepresentation of that reanalysis product in the SD ensemble. Large differences among the simulations constrained with the same reanalysis fields are also evident in $\Gamma_{\text{STRAT}}$ (Fig. 2e) and $\Gamma_{\text{NH}}$ (Fig. 2f), for which the SD ensemble spread is dominated by differences among the ERA-I and MERRA ensembles, respectively (Table 3, row 2). Note that this partly reflects the fact that more $\Gamma_{\text{STRAT}}$ output was available from ERA-I simulations (and more $\Gamma_{\text{NH}}$ output from MERRA simulations). Furthermore, among the MERRA-constrained simulations of $\Gamma_{\text{NH}}$ three of the simulations represented utilize similar models (e.g. WACCM 5hr, WACCM 50hr and CAM). Therefore, the particular details of the mean age differences discussed are likely sensitive to the choice of ensemble members and ensemble size.

## 4.2 Temporal Variability

### 4.2.1 Seasonal Cycle

The previous section showed that there are large differences in the climatological mean properties of various dynamical and transport fields among the SD simulations. Of these, the differences in the meridional winds are perhaps most surprising, given

that they were specified in all SD simulations. While it is true the meridional and zonal winds were nudged only indirectly in cases where nudging was applied to the divergence and vorticity model fields (Table 1), those four simulations cannot explain the intermodel spread exhibited among the larger SD ensemble. To explore this last point further we compare the temporal variability among the SD simulations, with respect to both seasonal and interannual timescales. In order to focus our analysis on the tropics and midlatitudes we restrict our analysis of temporal variability to spatial averages performed over latitudes between 60°S and 60°N, with the exception of the vertical velocities $\omega$ and $w^*$. For the latter variables, which change sign from positive to negative in the subtropics in both the troposphere and stratosphere, we perform averages over 30°S and 30°N. Our exclusion of latitudes outside the range 60°S-60°N is in order to avoid emphasizing the poles, where differences among the simulations may reflect large sensitivities to a few grid points and/or numerical instabilities. A discussion of the sensitivity of our results to choice of latitudinal bounds is presented in Section 5.

The seasonal cycle of U (Fig. 3a) agrees well among the SD simulations in both the upper and lower troposphere, with the exception of one outlier at 850 hPa. Closer inspection reveals that this particular ERA-I constrained simulation (i.e. UMUKCA-SD) corresponds closely to the free-running simulation produced using the same model, suggesting that its difference from the SD ensemble primarily reflected biases in its underlying free-running model (Supplementary Figure 3; Table 3, row 3). Comparisons of the seasonal cycle of the temperature, meridional winds and vertical winds also show generally good agreement among the SD simulations (Fig. 3b-d) in terms of the seasonal cycle phase (Fig. 4b-d,f-h, left) although the differences in phase for some variables are noticeably larger than for others. For example, the spread in $\tau_{\min}$ for $\omega_{850\mathrm{hPa}}$ and in in $\tau_{\max}$ for $V_{300\mathrm{hPa}}$ (Fig. 4b, left) is much larger than for the other fields. For the former case, this most likely reflects the fact that the seasonal cycle is not well defined over this latitudinal range and pressure level as there are two apparent minima occurring in both February and September (Fig. 3d, bottom). By comparison, for the latter case ($V_{300\mathrm{hPa}}$) the differences in seasonal cycle phase largely reflect differences among the reanalysis products, with MERRA exhibiting a much weaker seasonal cycle, relative to both ERA-I and JRA-55 (Fig. 3c, top) (Table 3, row 1).

While the seasonal cycle phases of U, V, T and $\omega$ are relatively well constrained by the SD simulations there are larger differences in the seasonal cycle amplitude (SCA) (Fig. 4, right panels). This is especially true for the meridional winds at both 300 hPa (Fig. 3c, Fig. 4b, right) and 850 hPa (Fig. 4f, right) and for the vertical winds, for which the SCA magnitude is anywhere between 0.3 and 1.2 the climatological mean value (Fig. 4d, right; Fig. 4h, right). Note that for the case of the former ($V_{300\mathrm{hPa}}$) part of this can be understood in terms of the use of different reanalysis products, with MERRA exhibiting a much weaker seasonal cycle in $V_{300\mathrm{hPa}}$, compared to both ERA-I and JRA-55. At the same time, however, Fig. 4b and Fig. 4f clearly show large differences among only the ERA-I (and MERRA) constrained simulations, indicating that both factors (i.e. different reanalysis products and implementation differences) contribute to the spread among the SD ensemble. Finally, note that the large normalized SCA values for the meridional wind fields reflect the fact that both $V_{300\mathrm{hPa}}$ and $V_{850\mathrm{hPa}}$ transition from positive to negative during the course of the annual cycle, which renders the annual climatological mean much smaller than the (unnormalized) SCA amplitude. For these two cases, therefore, the normalization of the SCA is somewhat less meaningful as a measure of seasonality, compared to the other variables.

Comparisons of the seasonal cycle of the TEM circulation and stratospheric tracers (Figure 5) also show generally good agreement in terms of the seasonal cycle phase among the SD simulations (Figure 6, left). The main exceptions are $N_2O$ at both 30 hPa and 80 hPa (Fig. 6d) and $O_3$ at 80 hPa (Fig. 6c, bottom), where $\tau_{min/max}$ varies widely across the simulations. As shown in Figure 5, this most likely reflects the fact that the seasonal cycle of these species is not well defined over this latitudinal and pressure range, indicating that care needs to be taken when interpreting $\tau_{min/max}$ since even subtle differences may manifest as large differences in seasonal cycle phase.

Similar to the tropospheric flow measures, the differences in SCA among the stratospheric transport and dynamical quantities are relatively larger, especially for $v^*_{80hPa}$ (Fig. 6a, bottom, right) and for $w^*_{30hPa}$ (Fig. 6b, top, right). The differences in SCA among the chemical and idealized tracers are also large (Fig. 6c-f, right) and, as in the troposphere, appear to be primarily associated with implementation differences and not to underlying differences among the reanalysis products. Furthermore, an additional comparison of the TEM and stratospheric tracer SD output with that from corresponding free-running simulations (not shown), reveals no systematic relationship between the SD ensemble biases and underlying free-running model biases (Table 3, row 3). Therefore, this indicates that the implementation of nudging is the largest source of spread in SCA for the stratospheric metrics considered here. Finally, given the fact that the seasonal cycle is not always well defined for all variables, we have checked the sensitivity of our calculations to the choice of latitudinal bounds over which the different fields were averaged before evaluating seasonal cycle phase and amplitude. A discussion of these sensitivities is presented at the end of Section 5.

### 4.2.2 Interannual Variability

We now extend our analysis to interannual timescales over the period 1980-2009. Deseasonalized timeseries of annual mean U, T, V, and $\omega$, averaged over 60°S to 60°N (30°S to 30°N for $\omega$), covary well among the SD simulations (Figure 7). Specifically, for U the average correlation coefficient among simulations in the SD ensemble ($\overline{\text{corr}}_{U,SD}$) is 0.97 at 300 hPa and 0.93 at 850 hPa (Table 4, Column 2). The correlations in zonal wind among simulations within each analysis ensemble are also high, consistently exceeding 0.93 (Table 4, Columns 3-5). Like the zonal winds, the temperature fields also covary well in the SD ensemble, with correlation coefficients of 0.95 (300 hPa) and 0.83 (850 hPa). Evaluating the covariability among the different analysis ensembles reveals that the somewhat poorer correlation values for T in the lower troposphere reflect differences among the ERA-I simulations, which have a correlation coefficient of 0.7 (Table 4). Closer inspection reveals that this is due to three of the ERA-I simulations (i.e. CHASER, IPSL and UMUKCA) and is consistent with the fact that the CHASER-SD simulation applied a much longer nudging timescale for T, compared to U and V (7 days versus 0.8 days) while UMUKCA was nudged to U, V and $\theta$, but not explicitly to T. The covariability in the meridional winds (Fig. 7c) and vertical winds (Fig 7d) is weaker than for the zonal winds and temperatures, although overall they are generally strong ($> 0.7$). In some cases these weaker correlations are related to differences in covariability among the reanalysis products, as for the case of $V_{300hPa}$, where the variability differs between MERRA and ERA-I, particularly over the period 1992-2002 (Fig. 7c, left; Table 3, row 3).

Moving next to the stratosphere we also find generally strong correlations among timeseries of $v^*$ and $w^*$, with values of $\overline{\text{corr}}_{v^*,SD}$ equal to 0.87 and 0.78 at 30 hPa and 80 hPa, respectively, and $\overline{\text{corr}}_{w^*,SD}$ equal to 0.93 and 0.92 (also at 30 hPa and 80

hPa) (Figure 8 a,b). The weaker correlation coefficient associated with $\overline{\text{corr}}_{v^*,\text{SD}}$ at 80 hPa appears to be associated with the use of different reanalysis products, especially during years 1994-2000, where the ERA-I constrained simulations exhibit a sharp decrease, not reflected either in the JRA-55 or MERRA simulations (Fig. 8a, right; Table 3, row 1). Among the constituents the correlations are relatively weaker although still generally strong, with positive correlations among simulations of ozone

($\overline{\text{corr}}_{O_3,\text{SD}}$ = 0.73 at 30 hPa and 0.90 at 80 hPa) and $N_2O$ ($\overline{\text{corr}}_{N_2O,\text{SD}}$ = 0.95 at 30 hPa and 0.99 at 80 hPa). (Note that the higher correlations in $N_2O$ partly reflect the underlying positive multi-decadal trend). The age correlations are also strong, all exceeding 0.81. When evaluating the covariability among the age tracers we did not include the results from the GEOS Replay and WACCM 5hr simulations since those tracers were integrated with initial conditions that were not spun up, consistent with the description in Orbe et al. (2017b), whose comparisons focused on 2000-2009 climatological means. Therefore, given that

those tracers do not equilibrate until the year $\sim$ 2000 we did not include them in our correlation analysis. Finally, as with our seasonal analysis, we have evaluated the sensitivity of our correlation analysis to the choice of spatial averaging; both sensitivity analyses are presented in section 5.

### 4.2.3 Trends

To conclude this section we now briefly comment on trends, as inferred by simply taking the linear fits of the timeseries shown

in Figures 7 and 8 over the period 1980-2009. For the tropospheric dynamical measures U, V, T and $\omega$, the trends exhibited by the SD simulations are in some cases in good agreement with the trends in the corresponding reanalyses (e.g. $U_{300}$, Fig. 9a, left). More generally, however, there is a large spread in the trends exhibited by the ERA-I ensemble, especially for the case of the meridional winds. As noted several times earlier this discrepancy is somewhat surprising given that V was explicitly constrained in all of the SD simulations.

In the stratosphere the spread in the trends is also large and, for the cases of $v^*$ and $w^*$, larger than the spread in the trends exhibited by the reanalyses themselves (Fig. 9 e-g). This latter point is consistent with Chrystanthou (see their Figure 11c) who showed that the trends in the tropical upward mass flux in nudged CCMI simulations generally did not match those from the reanalyses to which they were nudged. Similar behavior is exhibited by the age tracers (not shown). While the trends in the constituents (i.e. ozone and nitrous oxide) generally agree in sign (negative and positive, respectively) the source of this

agreement is likely driven by consistent variations in their sources and sinks, *not* by consistent underlying dynamical trends.

## 5   Free-Running versus Specified-Dynamics Simulations

### 5.1   Climatology

In the previous section we showed that certain aspects of the SD simulations (e.g. seasonal cycle phase, interannual variability) appeared to be much better constrained compared to others (e.g. climatological means, seasonal cycle amplitude), relative

to both the SD ensemble mean and the different reanalysis products. We now place these results in a broader context by comparing the SD simulations relative to free-running simulations produced *using the same underlying models*. To this end,

therefore, we focus only on the subset of the SD simulations for which modelers also submitted a corresponding free-running simulation (Column 4, Table 1), designated throughout as the SD* ensemble. Thus, in this section we focus on how well the SD* ensemble performs relative to FR ensemble, both of which consist of the same number $N$ of ensemble members. Note that for cases where multiple nudged simulations were submitted (e.g. WACCM 5-hr and 50-hr simulations) we only use one (in this case WACCM 50-hr) to ensure that both the SD* and FR ensembles have the same number of members.

Calculations of the root-mean-square (RMS) spread reveal interesting differences between the SD* and FR ensembles (Figure 10). Specifically, for a given variable $\chi$ the RMS spread for the $N$-member SD* multi-model ensemble is defined at each pressure level and latitude as follows: $\overline{\mathrm{rms}}_{\chi,\mathrm{SD}^*} \equiv \sqrt{\frac{1}{N}\sum_{i=1}^{N}(\chi(i) - \overline{\chi}_{\mathrm{SD}^*})^2}$. Similarly, $\overline{\mathrm{rms}}_{\chi,\mathrm{FR}}$ refers to the RMS spread averaged over the (also $N$-member) FR ensemble. Comparisons of $\overline{\mathrm{rms}}_{\mathrm{U,SD}^*/\mathrm{FR}}$ and $\overline{\mathrm{rms}}_{\mathrm{T,SD}^*/\mathrm{FR}}$ reveal that throughout the depth of the troposphere the zonal winds and temperatures are more consistent among the SD simulations, relative to the free-running models (Fig. 10a,c). By comparison, throughout the troposphere the values of $\overline{\mathrm{rms}}_{\mathrm{V,SD}^*}$ and $\overline{\mathrm{rms}}_{\mathrm{V,FR}}$ (Fig. 10b) are nearly identical, while, for the vertical winds, the spread among the SD* ensemble is systematically larger than among the FR ensemble by $\sim$20% (Fig. 10d). This suggests that nudging actually produces *larger* intermodel differences in the vertical winds, relative to those associated with underlying free-running model biases. While it is true that the vertical component of the wind field is not a prognostic variable (and, hence, not directly nudged) the larger spread in $\omega$ among the SD ensemble is, at the very least, surprising.

RMS spread comparisons of the stratospheric circulation and transport measures reveal a similar story, with similar values of $v^*$, $w^*$, and $O_3$ (Fig. 10e,f,g) among both the SD* and FR ensembles. In the middle and upper stratosphere the RMS spread is consistently greater in the SD ensemble for both $N_2O$ (Fig. 9h) and $\Gamma_{\mathrm{STRAT}}$ (Fig. 9i). Interestingly, the RMS comparisons of the age tracers do not produce a consistent story above 50 hPa which, upon first glance, seems contradictory. However, as discussed earlier, this is because the SD ensembles for $\Gamma_{\mathrm{STRAT}}$ and $\Gamma_{\mathrm{NH}}$ consist of very different models. Specifically, the SD models included in the comparisons of $\Gamma_{\mathrm{NH}}$ include three MERRA-constrained simulations performed using models from the same modeling center (WACCM 5hr, WACCM 50hr, CAM). Therefore, the smaller RMS spread for that tracer in the SD* ensemble needs to be interpreted with caution, as it reflects similarities among three simulations produced using the same (or very similar) underlying model.

Finally, as discussed earlier our decision to average over 60°S and 60°N may mask potentially interesting regions of RMS spread that are smoothed out upon averaging and/or may raise concerns about the robustness of our conclusions. Therefore, we also compared the pressure and meridional distributions of the RMS spread between the SD* and FR ensembles (Supplementary Figures 4 and 5). Overall, the RMS patterns reflect the underlying structure of the model field such that regions of strong spatial gradients and/or reversals in sign tend to align with regions where the RMS spread is larger. This applies to both SD* and FR ensembles, with the exception of U, for which the RMS values in the SD* ensemble are negligible throughout the troposphere. While these patterns are interesting, they are more or less symmetric about the equator, indicating that the use of a 60°S and 60°N averaging operator does not pose any obvious concerns regarding robustness of our results. Furthermore, for certain variables the RMS spread exhibits strong vertical gradients (e.g. T, V) that support our use of 300 hPa and 850 hPa and 30 hPa and 80 hPa as representative pressure levels in the troposphere and stratosphere, respectively.

## 5.2 Variability

Comparisons of the seasonal cycle among the SD$^*$ and FR ensembles (Figure 11) show that seasonal cycle phase is generally more consistent among the SD$^*$ simulations, compared to the FR simulations, although there are cases where the differences in phase spread among the ensembles is similar (e.g. $\omega$ and V in Fig. 11b,d,f,h, left). The seasonal cycle amplitude (SCA) is also somewhat better constrained in the SD$^*$ ensemble, at least for U and T. However, there are large differences in SCA amplitude in the meridional and vertical winds, evident in both the lower and upper troposphere (Fig. 11b,d,f,h, right).

The seasonal cycle phase of the TEM and transport circulations appears to be slightly better constrained among the SD$^*$ versus FR ensembles (Fig. 12, left). As with the other variables, however, the seasonal cycle amplitude is, by comparison, less well constrained in both SD$^*$ and FR ensembles. Specifically, at 80 hPa the seasonal cycle amplitude differences among the SD$^*$ runs are larger than among the FR models for the cases of $v^*$ at 80 hPa (Fig. 12a, bottom), $w^*$ at 80 hPa (Fig. 12b, bottom), $O_3$ at 80 hPa (Fig. 12c, bottom), $N_2O$ at 30 hPa (Fig. 12d, top), and $\Gamma_{\mathrm{STRAT}}$ at 30 hPa (Fig. 12e, top). Overall, therefore, upon comparing ensembles of equal sizes, we conclude that, while the seasonal cycle phase is slightly better constrained in the SD$^*$ ensemble, the amplitude is not.

As with our analysis of the RMS spread, we have also checked the sensitivity of our seasonal cycle calculations to the choice of latitudinal averaging bounds. As indicated earlier in Section 4, in a few cases the seasonal cycle was either too small in amplitude or not characterized by a unique maximum/minimum, which raised questions about the appropriateness of the SCA and $\tau_{\mathrm{min/max}}$ diagnostics. Therefore, in addition to that analysis we have evaluated the correlation of the seasonal cycle at each grid point for each member of both the SD$^*$ and FR ensembles, relative to the SD$^*$ and FR ensembles averages, respectively. Supplementary Figures 6 and 7 show that for U, V T and $\omega$ the SD ensemble shows overall high correlations over all levels and latitudes, except for the tropical mid-troposphere between 300-700 hPa for V, where the meridional winds transition sign from mean southerly/northerly flow; in this latter case the low correlation coefficients therefore most likely reflect differences between small numbers. For the TEM variables there is also interesting spatial structure in the correlations of $v^*$ and $w^*$ for the SD$^*$ ensemble, with relatively lower correlations in the lower and middle stratosphere, and for $N_2O$, with relatively lower correlations over the NH middle and high latitudes for both SD and FR ensembles. Overall, however, the spatial patterns of correlation coefficients for all variables are more or less symmetric about the equator and span much of the subtropics and extratropics within our latitudinal averaging bounds. Therefore, while this spatial structure is interesting on a case-by-case basis, we feel that the use of 60°S to 60°N (30°S to 30°N for $\omega$ and $w^*$) latitude averaging bounds is appropriate for synthesizing our results and does not hinder the robustness of our conclusions.

Comparisons of the RMS spread between the FR and SD$^*$ ensembles (Table 4, last two columns) reveal that the SD simulations nearly always exhibit much more consistent interannual variability, compared to their free-running counterparts. This is particularly clear for the meridional winds in the upper troposphere (300 hPa), where $\overline{\mathrm{corr}}_{\mathrm{V,SD^*}} = 0.75$, compared to $\overline{\mathrm{corr}}_{\mathrm{V,FR}}$ = 0.45. Similarly, the vertical velocity interannual variability is much better constrained in the SD$^*$ ensemble, with $\overline{\mathrm{corr}}_{\omega,\mathrm{SD^*}}$ = 0.86 compared to $\overline{\mathrm{corr}}_{\omega,\mathrm{FR}}$ = 0.35 at 300 hPa. The TEM circulation also covaries better among the SD simulations as do ozone variations at both 30 hPa and 80 hPa, with correlation coefficients that are about 0.2 larger than for the FR simulations.

Finally, the trends simulated by the SD* ensemble show a similar – if not larger – disagreement compared to the trends simulated by their corresponding FR models (Figure 13). While for the cases of U, V and T (Fig. 13 a-c) the spread in the SD* simulated trends is somewhat smaller this does not apply generally, especially for the cases of $v^*$, $w^*$ and the constituents (Fig. 13. e-g). Chrysanthou et al. (2019) came to a similar conclusion with respect to the tropical upward mass flux, which they showed exhibited larger trend discrepancies, compared to the free-running simulations (compare panel c in their Figures 10 and 11). Large trend discrepancies in the tropical upward mass flux were also exhibited by the SD simulations, compared to the reanalyses (compare their Figure 13 and Supplementary Figure 17).

## 5.3 Dynamical Consistency

Whereas in the previous sections we evaluated the SD simulations in terms of their representation of individual fields here we briefly examine the dynamical consistency of the large-scale circulation. Given the surprising differences in the tropospheric meridional winds we restrict our attention to the tropical mean meridional circulation and, in particular, to the Hadley Cell (HC). Waugh et al. (2018) compared a broad range of lower and upper tropospheric measures of the HC and found that the strongest relationships occurred between the HC edge based on the near-surface zonal winds (hereafter denoted as UAS) and the HC edge based on the meridional mass streamfunction (hereafter PSI) among both reanalysis and free-running models from CMIP5 (Taylor et al. (2012)). Furthermore, they showed that strong correlations between UAS and PSI occur not only on interannual timescales but also in terms of their trends and forced responses to global warming.

Figure 14 compares UAS and PSI among the SD* and FR ensembles. Specifically, UAS corresponds to the first subtropical latitude where the near-surface zonal wind changes from negative to positive. By comparison, PSI corresponds to the zero-crossing of the meridional mass streamfunction ($\Psi$) at 500 hPa, where $\Psi$ was calculated as the vertical integral of the meridional component of the zonal mean wind using the TropD software package from Adam et al. (2018). The comparisons of UAS and PSI first show that UAS is very well constrained in both hemispheres among the SD* simulations, exhibiting a spread that is much smaller ($\sim$1-2$^\circ$), compared to their corresponding FR simulations (up to $\sim 10^\circ$) (Fig. 14a). By comparison, the spread in PSI is much larger, especially during boreal summer in the NH where PSI differs by $\sim 25^\circ$ compared to a much smaller range among the FR simulations ($\sim 10^\circ$) (Fig. 14b). More importantly, the relationship between UAS and PSI is entirely different between the SD and FR ensembles. That is, consistent with Waugh et al. (2018), the FR simulations exhibit a strong positive relationship between UAS and PSI, especially in the SH, such that a more poleward UAS is associated with a more poleward PSI. This relationship is not demonstrated by the SD simulations, indicating that the meridional and zonal components of the flow are not dynamically consistent in that ensemble of runs, similar to the results presented in Davis and Davis (2018), although their focus was on the actual reanalyses fields (not nudged simulations).

## 6 Conclusions

The main goal of this study has been to document how the REF-C1SD experiment was implemented across the Chemistry-Climate Model Initiative (CCMI) models, since this information is not available in the published literature. While some of the

information described here is addressed in Supplementary Table 30 of Morgenstern et al. (2017), we have included a more complete description, based on information solicited from individual modeling groups in the form of a community survey. Furthermore, we have also used this opportunity to present a more rigorous evaluation of several dynamical and transport fields that were provided as output but were only briefly discussed in Orbe et al. (2018) and Yang et al (2019). Our analysis has distinguished between how well the specified-dynamics (SD) simulations represent climatological zonal mean distributions versus temporal variability, with respect to the entire SD ensemble, reanalysis products as well as free-running simulations produced using the same underlying models. Our conclusions are summarized as follows:

– Comparisons of the climatological annually and zonally averaged zonal winds and temperatures show good agreement in the troposphere among the SD simulations and with respect to the reanalysis fields. By comparison, the differences in the meridional winds and vertical winds are much larger ($\sim$ 30-40%) and are related both to the use of different reanalysis products and to differences in implementation. In the stratosphere, the spread in the climatological Transformed Eulerian Mean (TEM) and transport circulations among the SD simulations is also large (approaching $\sim$ 100%) and is primarily related to differences in implementation.

– For most variables (both tropospheric and stratospheric) there is good agreement ($<$ 20% spread) in terms of the phase of the seasonal cycle; by comparison, the seasonal cycle amplitude (SCA) exhibits much larger differences ($\sim$ 50%). On interannual timescales, the SD simulations exhibit good covariability (correlation coefficients $>$ 0.7) for nearly all fields.

– Overall, the spread in both the mean climatological distributions and SCA among the SD simulations cannot be attributed solely to the use of different reanalysis products. While in some cases (e.g. $V_{300hPa}$) the differences among the reanalysis products are large, in general the SD spread is much larger.

– For most variables the SD simulations perform similarly – and in several cases (e.g. meridional winds, TEM circulation) worse than – free-running simulations *produced using the same models* in terms of their climatological mean values and seasonal cycle amplitudes. By comparison, the SD simulations consistently exhibit superior covariability on interannual timescales for nearly all variables analyzed here, although their trends differ substantially both with respect to each other and compared to their corresponding reanalyses, consistent with findings from previous studies.

– Interestingly, the relationship between the meridional and zonal components of the flow is fundamentally different between the SD simulations and the FR simulations. Unlike the free-running simulations, the specified-dynamics simulations do not exhibit a strong correlation between indices of the Hadley Cell derived separately from the zonal versus meridional winds. This reveals that different components of the flow are not dynamically consistent in all of the SD simulations.

We have shown that there are large differences in how SD simulations represent the mean climatological distributions and seasonal cycle phases of various tropospheric and stratospheric flow and transport measures. The differences in the meridional winds are particularly surprising, given that all simulations were explicitly constrained to meridional winds derived from the

analysis fields. At the same time, we also showed that the SD simulations exhibit much better covariability on interannual timescales, relative to free-running simulations using the same underlying models. Note that, upon testing the sensitivity of our analysis to choice of metrics, we found that for a few variables and locations the phase of the seasonal cycle was not well defined (e.g. $N_2O$), in which cases the spread in $\tau_{min/max}$ may be less meaningful. However, these cases were anomalies and, after redoing our analysis in terms of similar but distinct metrics (e.g. correlation of the seasonal cycle versus SCA), we found qualitatively similar results supporting our original conclusions. In addition, we also found that our main conclusions were robust to how our calculations were performed, specifically with respect to the choice of both latitudinal averaging bounds and pressure levels.

Overall, our analysis suggests that studies using SD simulations should exhibit strong caution when inferring the influence of dynamics on tracers. More precisely, our results indicate that studies relating large-scale dynamics to atmospheric transport would be most justified in using SD simulations to examine science questions related to interannual variability; by comparison, studies would be less justified to address questions hinging on credible representations of the seasonal cycle amplitude, trends or the overall magnitude of the large scale flow. The lack of dynamical consistency exhibited by some SD simulations, at least with respect to the tropospheric subtropical flow, also raises concerns that may complicate the interpretation of results using nudged simulations to study the influence of atmospheric dynamics on composition. Overall, several of our findings are consistent with the analysis in Chrysanthou et al. (2019), who provided a thorough comparison of the stratospheric residual mean circulation among the REF-C1SD simulations, but their analysis does not extend to stratospheric trace gases (or the troposphere). In the spirit of providing a review of the REF-C1SD experiment our focus here has been broader in scope.

An important conclusion from our analysis is that the differences among the SD simulations are not primarily driven by differences between the reanalysis fields. To this end we have attributed the SD ensemble spread primarily to "differences in implementation." It is important to clarify, however, that by "implementation" we refer to *both* the departures from the analysis fields associated with nudging as well as biases associated with the underlying free-running models (Table 3, rows 2 and 3). Therefore, for those fields for which it was shown that outlier SD simulations closely tracked their corresponding free-running simulations, our conclusion is that the SD ensemble spread primarily reflects biases in the underlying free-running models (e.g. $T_{850hPa}$). For other cases, however, in which the SD ensemble spread was shown to be larger than in the FR ensemble (e.g. $w^*, \omega, N_2O$), we conclude that the act of nudging actually produces *larger* divergence among the models than would be expected solely due to underlying differences in model formulation. Furthermore, as discussed in Section 5, while the SD ensemble included results from three chemical transport models (CTMs), the majority of simulations considered here were performing using nudged chemistry climate models (CCMs). While we could not identify clear CTM vs. nudged differences in our analysis in Section 5 future studies should focus on more systematically comparing the performance of nudged simulations not only relative to free-running simulations, as examined here, but also relative to offline CTMs.

One final caveat of our analysis is that we have only compared the resolved large-scale flow. Therefore, when interpreting the transport differences, reflected in both the idealized and chemical tracers, one must also consider differences in transport related to sub-grid scale processes (e.g. parameterized convection, vertical diffusion) (Orbe et al. (2017a), Orbe et al. (2018)). In particular, Orbe et al. (2018) showed that the parameterized convection differences in the troposphere are even larger among

the REF-C1SD simulations, relative to the REF-C1 ensemble, especially in the tropics. Our analysis here, therefore, has aimed solely at providing a more detailed description of the large-scale flow representation in the SD ensemble, compared to the briefer discussions presented in earlier works. Finally, the second assumption that we have made is that any inconsistencies related to the use of different advection schemes used for simulating the flow and tracers are small, relative to transport differences arising in response to how nudging is implemented among the various simulations. However, as noted in Morgenstern et al. (2017), only one of the CCMI models considered here (UMUKCA) uses different schemes for the advection of chemical versus physical tracers (e.g. momentum, heat), indicating that our assumption is valid in the context of the larger SD ensemble.

The authors declare that they have no conflict of interest.

*Data availability.* The majority of the data of CCMI Phase 1 used in this study can be obtained through the British Atmospheric Data Centre (BADC) archive (ftp://ftp.ceda.ac.uk). For instructions for access to this archive see http://blogs.reading.ac.uk/ccmi/badc-data-access. Output from the WACCM and CAM simulations was obtained from the NCAR Earth System Grid portal (https://www.earthsystemgrid.org/). In addition to the CCMI and NCAR data we obtained MERRA U, V, T from the assimilated (ASM) (not analysis) collections, located on the NASA machines maintained by the NASA Center for Climate Simulation (NCCS).The ERA-I and JRA-55 U, V, T and $\omega$ fields were downloaded from the Research Data Archive at the National Center for Atmospheric Research, Computational and Information Systems Laboratory (Japan Meteorological Agency/Japan (2013); European Centre for Medium-Range Weather Forecasts (2012)). The TEM velocities ($v^*$ and $\omega^*$) from the SPARC Reanalysis Intercomparison Project (S-RIP) dataset were downloaded from https://catalogue.ceda.ac.uk/uuid/b241a7f536a244749662360bd7839312.

.

*Author contributions.* CO performed the analysis of the CCMI simulations. CO, DP, DWW and HY conceived of the analysis and provided significant guidance about the manuscript and figures presented. PJ, DEK, BJ, VJ, MD, NLA, ATA, MPC, SD, WF and SB performed the CCMI simulations in support of the CCMI Phase 1 and provided constructive feedback on the manuscript.

*Acknowledgements.* We thank the Centre for Environmental Data Analysis (CEDA) for hosting the CCMI data archive. We acknowledge the modeling groups for making their simulations available for this analysis, and the joint WCRP SPARC/IGAC Chemistry-Climate Model Initiative (CCMI) for organizing and coordinating this model data analysis activity. In addition, C.O. wants to thank the high-performance computing resources provided by the NASA Center for Climate Simulation (NCCS) as well as support from the NASA Modeling, Analysis and Prediction (MAP) program. D.W. acknowledges support from NSF grant AGS-1403676 and NASA grant NNX14AP58G. The EMAC simulations have been performed at the German Climate Computing Centre (DKRZ) through support from the Bundesministerium für

Bildung und Forschung (BMBF). DKRZ and its scientific steering committee are gratefully acknowledged for providing the HPC and data archiving resources for this consortial project ESCiMo (Earth System Chemistry integrated Modeling). UMUKCA model integrations were performed using the ARCHER UK National Supercomputing Service and the MONSooN system, a collaborative facility supplied under the Joint Weather and Climate Research Programme, which is a strategic partnership between the UK Met Office and the Natural Environment Research Council. TOMCAT is supported by National Centre for Atmospheric Science (NCAS).

## Appendix A: Description of Individual REF-C1SD Simulations

### A1 CESM1 CAM4-chem and CESM1 WACCM

The CAM4-chem and WACCM 5hr(50hr) REF-C1SD simulations are nudged to MERRA 6-hourly instantaneous analysis fields, applying a mass conserving interpolation of the MERRA fields (1/2° latitude by 2/3° longitude) to the models' horizontal grids (1.9 ° latitude x 2.5 ° longitude). Nudged meteorological fields include both three-dimensional fields (T, U, V) as well as the two-dimensional fields PS, TAUX, TAUY, SHFLX, LHFLX (surface pressure, surface stress, latent heat flux, sensible heat flux). Note that water vapor is derived in the model. Nudging occurs over a 30 minute timestep and is applied linearly (in pressure) from the surface to 50 km, above which the simulation is fully fully free-running. The nudging relaxation time constant $\tau$ is spatially constant and set to 50 hrs$^{-1}$ (5 hrs for WACCM 5hr). Spectral nudging is not used.

In addition to being forced by SSTs and SICs from HADISST, the CAM4-chem and WACCM REF-C1SD simulations use PHIS (topography) from MERRA. Drifts in surface pressure that are generated by nudging are corrected for at every advection step (or every 30 minutes/8 = 3.75 minutes). The convective mass fluxes are not taken from MERRA, but rather, derived from CAM.4 column physics, which represents convection using the parameterizations of Zhang and McFarlane (1995) and Hack (1994) for deep and shallow convection, respectively.

Reference: Lamarque et al. (2013)

### A2 CCSRNIES MIROC3.2

The CCSRNIES MIROC3.2 REF-C1SD simulation is nudged to six-hourly ERA-I instantaneous fields and forced with prescribed HADISST1 boundary conditions. The analysis fields are linearly interpolated temporally to the model time and spatially interpolated linearly in the horizontal and linearly with respect to log-pressure levels in the vertical. The nudged three-dimensional meteorological fields are T, U, and V using a nudging timestep equal to the model timestep for dynamics and nudging is applied from the surface to 1 hPa. Above 1 hPa U and T are nudged to zonal mean fields obtained from CIRA, with no representation of year-to-year variability. The nudging relaxation time constant $\tau$ is set to a value of 24 hours$^{-1}$ throughout the domain and spectral nudging is not used. To correct for surface pressure drifts a pressure correction is applied. Convective mass fluxes are recalculated online using the parameterization described in Arakawa and Schubert (1974).

Reference: Akiyoshi et al. (2016)

## A3  HadGEM3-ES

The HadGEM3-ES REF-C1SD simulation is nudged to ERA-I six-hourly instantaneous fields and forced with HADISST1 SSTs. The analysis fields are linearly interpolated temporally to the model timestep; spatially, bilinear interpolation is applied in the horizontal and linear (in log pressure) interpolation in the vertical. The three-dimensional fields T, U, and V are nudged every model timestep (i.e. every 20 minutes) to a value interpolated between the instantaneous fields valid at the previous and next six-hour slot (e.g. between data for 00:00 GMT and 06:00 GMT). While nudging is uniform at all levels between 2.5 km and 50 km there is transition from 0 to full-strength nudging over the top and bottom levels of the nudging domain. Neither spectral nudging nor a pressure correction for surface pressure drifts is used. Moist convection is parameterized using Walters et al. (2014).

Reference: Hardiman et al. (2017)

## A4  GFDL-AM3

The GFDL-AM3 REF-C1SD simulation is nudged to NCEP/NCAR T62 six-hourly instantaneous fields and constrained with HADISST2 SSTs. The NCEP reanalysis fields, which are at T62 horizontal resolution, are interpolated to the C48 native model cubed sphere grid ($\sim$200 km by $\sim$200 km). Only the horizontal winds U and V are nudged using a nudging timestep equal to the model time step (i.e. every 30 minutes). Nudging is pressure-dependent between the surface and 10 hPa, with $\tau = 6$ hours in the surface level and weakening linearly (in pressure) to 60 hours at 100 hPa and 600 hours at 10 hPa. Neither spectral nudging nor a pressure correction for surface pressure drifts is used. The convective mass fluxes are calculated online using the parameterization described in Donner et al. (2011).

References: M. Lin et al. (2012a, 2014, 2015a,b)

## A5  TOMCAT CTM

The TOMCAT CTM simulation is constrained using ERA-I six-hourly instantaneous analysis fields which are interpolated linearly in time to the model timestep (1 hour). Horizontally, the vorticity, divergence, surface pressure and temperature fields are read in as spectral coefficients and a spectral transform is applied which averages those fields onto the model horizontal grid. Kinematic velocities are used in the vertical and are determined by reading the vorticity in every six hours and using that to compute the vertical mass flux, which is distributed over the model levels in order to conserve total mass advection. The TOMCAT simulation is among a subset of the REF-C1SD simulations that are not forced with observed boundary conditions but, rather, SSTs and SICs taken from an old NCAR CCM-II model. Convective mass fluxes are calculated online using the moist convective mass flux scheme of Tiedtke (1989). The model has parameterizations for sub-grid scale tracer transport by convection (Stockwell and Chipperfield (1999); Feng et al. (2011)) and boundary layer mixing (Holtslag and Boville (1993)).

Reference: Chipperfield et al. (2006)

## A6 IPSL LMDz-REPROBUS

The IPSL LMDz-REPROBUS REF-C1SD nudged simulation is constrained with ERA-I six-hourly instantaneous fields and forced with SSTs and SICs from the AMIP II dataset. The nudged fields U, V and T are linearly interpolated spatially to the model grid and linear interpolated temporally to the model timestep (1/2 hour). Nudging occurs everywhere and operates on a
675 timescale ranging up to 3 hours, depending on altitude. Neither spectral nudging nor a pressure correction for surface pressure drifts is used. Convective mass fluxes are calculated online using the convection scheme of Dufresnes et al. (2013).

References: Marchand et al. (2012), Szopa et al. (2013) and Dufresne et al. (2013)

## A7 CMAM-SD

The CMAM REF-C1SD simulation is nudged to ERA-I six-hourly instantaneous fields and forced with HadISST1 boundary conditions. The three-dimensional meteorological fields are vorticity, divergence and temperature and are linearly interpolated vertically to the model grid and linear interpolated temporally to the model timestep (7.5 minutes). Spectral nudging is applied in the horizontal for all waves up to T21 (nudged with equal strength). While no interpolation is required in the horizontal, vertically, the analysis fields are linearly interpolated to the model levels. Nudging occurs at every model timestep using a
timescale $\tau = 24$ hours for all model levels up to 1 hPa, above which the model is free-running. No pressure correction is applied to correct for surface pressure drifts. The convective mass fluxes are recalculated online and parameterized using Zhang and McFarlane (1995).

References: Jonsson et al. (2004) and Scinocca et al. (2008)

## A8 UMUKCA

The UMUKCA REF-C1SD simulation is nudged to ERA-I six-hourly instantaneous fields and forced with HadISST SSTs and SICs. The three-dimensional nudged meteorological fields are U, V, and $\theta$ and are linearly interpolated temporaly to the model timestep (20 minutes) and spatially to the $\sim2.5°$ latitude by $\sim3.75°$ longitude model grid. Nudging is applied at every model timestep using a nudging e-folding timescale $\tau = 6$ hours over both the free troposphere and the stratosphere, although
the nudging strength equals 1% that in the troposphere, as diagnosed using the model's tropopause height. Neither spectral nudging nor a pressure correction is applied for to correct for surface pressure drifts. Convective mass fluxes are calculated online and parameterized as described in Hewitt et al. (2011).

References: Morgenstern et al. (2009) and Bednarz et al. (2016)

## A9 MOCAGE CTM

The MOCAGE CTM is driven with ERA-I six-hourly instantaneous fields, which are interpolated onto the regular 2° latitude by 2° longitude MOCAGE model grid and linearly interpolated in time every model timestep. In addition to the horizontal winds U and V, T and specific humidity (Q) are also derived from ERA-I. Kinematic vertical velocities are recomputed by the CTM to ensure that the mass flux is conservative on the CTM grid and are calculated at the same temporal frequency as the ERA winds. Convective mass fluxes are not taken directly from ERA-I but, rather, are re-computed from ERA-I large-scale fields based on the approach described in Bechtold (2001).

References: Josse et al. (2004) and Guth et al. (2016)

## A10 EMAC

The EMAC REF-C1SD simulation is nudged to ERA-I six-hourly instantaneous fields that are linearly interpolated to the model timestep and spatially interpolated to the T42 resolution of the model. Nudging is applied in spectral space with low normal mode insertion to the divergence, vorticity, temperature, and (the logarithm of) surface pressure fields. The nudging strength varies with pressure while the relaxation time constant $\tau$ varies among the meteorological fields, with $\tau = 6, 24, 24$ and 48 hours for the vorticity, surface pressure, temperature and divergence fields, respectively. SSTs and SICs for the free-running simulations are taken from the HadISST product and no surface pressure and no surface pressure correction is applied. For the SD setup SSTs and SICs (both 12 hourly) are taken from ERA-Interim, consistent with the data used for the Newtonian relaxation (nudging). Convection is parameterized as described in Tiedtke (1989) and Nordeng (1994). Both low and high vertical resolution model simulations (47 and 90 levels) were used to perform the REF-C1SD hindcast simulation. Note that EMAC provided two sets of SD simulations on the BADC archive, although here we only include the results from the pair of 47- and 90-level simulations in which temperatures were nudged to the entire analysis field in which global mean temperature is retained. A second pair of simulations, not considered here, was also provided, in which only temperature anomalies (with respect to the global spatial mean) were nudged.

Reference: Jöckel et al. (2016)

## A11 NASA GEOS-Replay

The NASA REF-C1SD simulation is produced using the Goddard Earth Observing System Version 5.0 (GEOS-5) model (Rienecker et al. (2008)) and the "replay" approach, which involves reading in MERRA fields and recomputing the analysis increments, which are applied as a forcing to the meteorology at every model time step (Orbe et al. (2017b)). The GEOS replay framework is similar to the standard GEOS data assimilation procedure in the sense that it uses the same Incremental Analysis Update (IAU) technique that is used to apply the analysis as a correction to the background state (Bloom et al. (1996)). Replay simulations, performed here at a C48 cubed sphere (or approximately 2° latitude by 2.5° longitude) horizontal resolution, are

constrained to MERRA zonal and meridional winds, temperature and surface pressure, while all other dynamical variables and physics are recalculated online; thus, unlike in a CTM, the parameterized convective mass fluxes are recalculated on-line using the Relaxed-Arakawa Schubert convective scheme (Moorthi and Suarez (1992)). SSTs and SICs are derived from the weekly one-degree sea surface temperature product of Reynolds et al. (2000), which is linearly interpolated to each model time step.

This is the same simulation as the "RAna" simulation examined in Orbe et al. (2017b). Note that this simulation is constrained with the MERRA analysis *not* the assimilated fields, consistent with its original implementation as in other studies (e.g. Colarco et al. (2010), Strode et al. (2015)). As discussed in Orbe et al. (2017b) GEOS replay simulations produce different stratospheric transport properties depending on whether the MERRA assimilated or analysis fields are used, with simulations constrained with the assimilated fields exhibiting stratospheric mean age values that are more consistent with observations.

Reference: Orbe et al. (2017b)

## A12    NASA GMI-CTM

The second NASA REF-C1SD simulation is produced using the Global Modeling Initiative Chemical Transport Model (Strahan et al. 2007, 2016). The GMI-CTM is driven with MERRA three-hourly time-averaged assimilated winds, including both the horizontal winds as well as kinematic vertical velocities. Tropospheric water vapor is also input from MERRA, while stratospheric water is assumed to be 3 ppm at the tropical tropopause and has a source from methane oxidation. (As such stratospheric water vapor in the GMI-CTM should not be used as a transport tracer). While there is no temporal interpolation between the meteorological field updates the assimilated fields are horizontally regridded to the GMI horizontal grid ($2°$ degrees latitude by $2°$ longitude) to ensure mass conservation. By comparison, there is no interpolatation in the vertical since both the CTM and the MERRA assimilated fields share the same vertical grid. SSTs and SICs are derived from the weekly one-degree sea surface temperature product of Reynolds et al. (2000), which is linearly interpolated to each model time step. Convective mass fluxes are taken from MERRA (interpolated from the native $0.5°$ latitude to $2.5°$ longitude horizontal grid) and, therefore, reflect the Relaxed Arakawa-Schubert convective parameterization used in GEOS-5, as detailed in Moorthi and Suarez (1992).

References: Strahan et al. (2007, 2016)

## A13    MRI-ESM1r1

The MRI-ESM1r1 REF-C1SD simulation is nudged to the JRA-55 6-hourly instantaneous reanalysis fields which are linearly interpolated spatially from the native analysis grid ($1.25°$ latitude by $1.25°$ longitude, 37 layers between 1000 hPa and 1 hPa) to the model grid. Temporal interpolation is also linear and the nudging time interval is set to the model time step of 30 minutes. Similar to the other model simulations the nudged meteorological fields are T, U, and V (PHIS (topography) is taken from MERRA). Nudging is applied using a constant timescale $\tau$ = 24 hours between 870 hPa and 40 hPa whereas above 40 hPa the nudging timescale increases with height as follows: $\tau = \tau_0 * (1-\log(p/40))/\log(1/40)$. Spectral nudging is not used. The

765 simulation is forced with HADISST V1.1 SSTs and surface pressure drifts are not corrected. The parameterized convective mass fluxes are computed online using the parameterization as described in Yoshimura et al. (2015).

References: Deushi and Shibata (2011) and Yukimoto et al. (2011)

### A14 Chaser-MIROC-ESM

The Chaser-MIROC-ESM REF-C1SD simulation is nudged to the ERA-I reanalysis fields U, V, and T using a constant timescale $\tau = 0.8$ days for the horizontal winds and $\tau = 7$ days for temperature. Nudging is applied uniformly between 10 hPa and 900 hPa. Spectral nudging is not used and the simulation is forced with HADISST SSTs. The parameterized convective mass fluxes are computed online using both Arakawa and Schubert (1974) and cloud base mass flux from Pan and Randall (1998).

Reference: Watanabe et al. (2011)

### A15 CNRM-CM5-3

The CNRM-CM5-3 REF-C1SD simulation is nudged to ERA-I six hourly instantaneous fields U, V, T and Q over all model levels, with a transition zone from the surface over the last five model levels. The nudging strength is spatially uniform and for 780 all variables equal to $\tau = 5$ hours. SSTs and SICs are taken from the HadISST product and no surface pressure correction is applied. Convection is parameterized as described in Bougeault (1985).

Reference: Douville (2009)

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

# Meridional Profiles at 300 mb (top) and 850 mb (bottom)

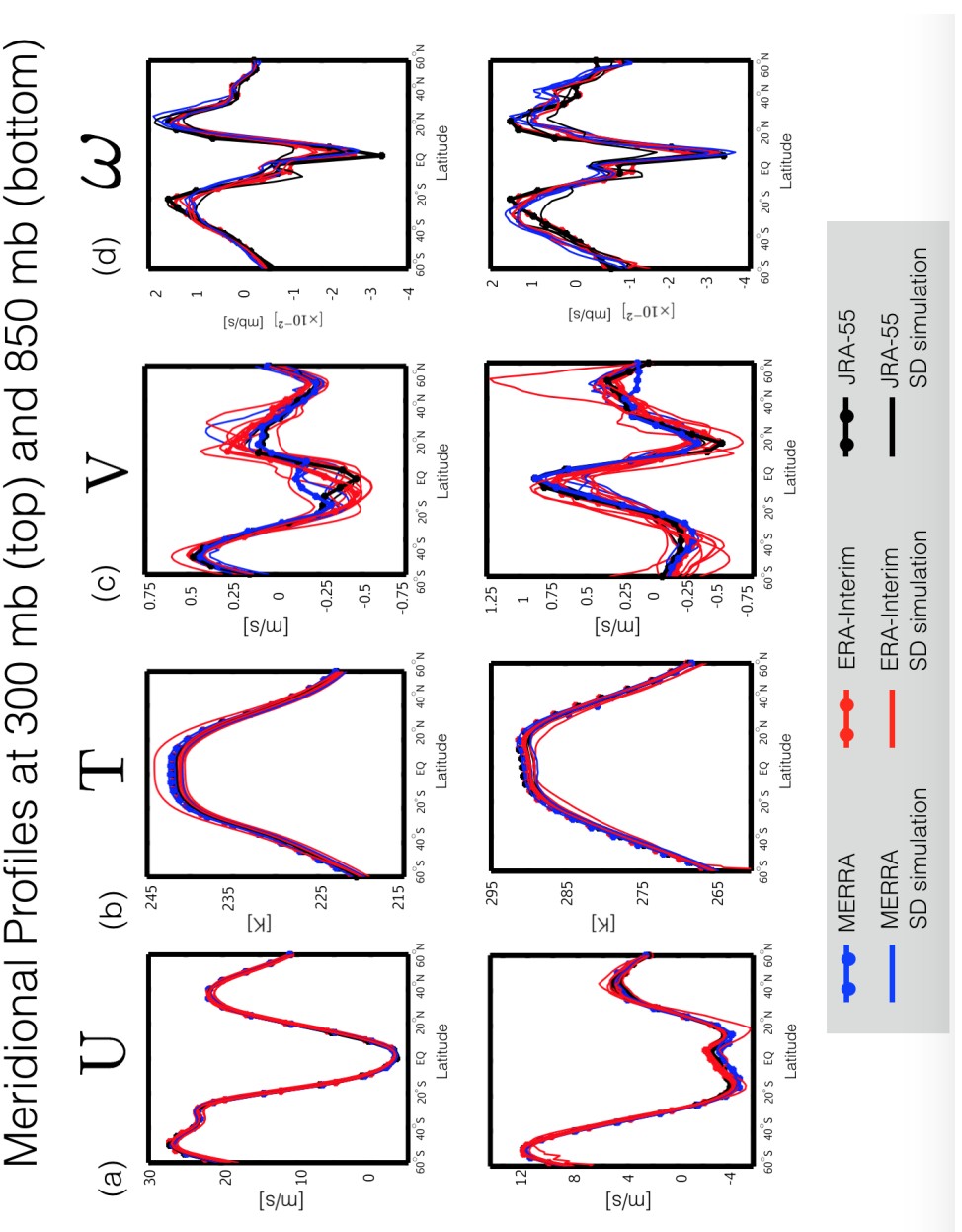

**Figure 1.** Meridional profiles of the 2000-2009 climatological annual mean (a) zonal mean zonal wind (U) (b) zonal mean temperature (T) (c) zonal mean meridional wind and (d) zonal mean pressure velocity ($\omega$) at 300 hPa (top panels) and 850 hPa (bottom panels). Red/blue/black lines correspond to SD simulations constrained with ERA-Interim/MERRA/JRA-55 reanalysis fields. Red/blue/black dotted lines correspond to the raw ERA-Interim/MERRA/JRA-55 meteorological fields. Note that MERRA assimilated fields are shown, for which $\omega$ is not available.

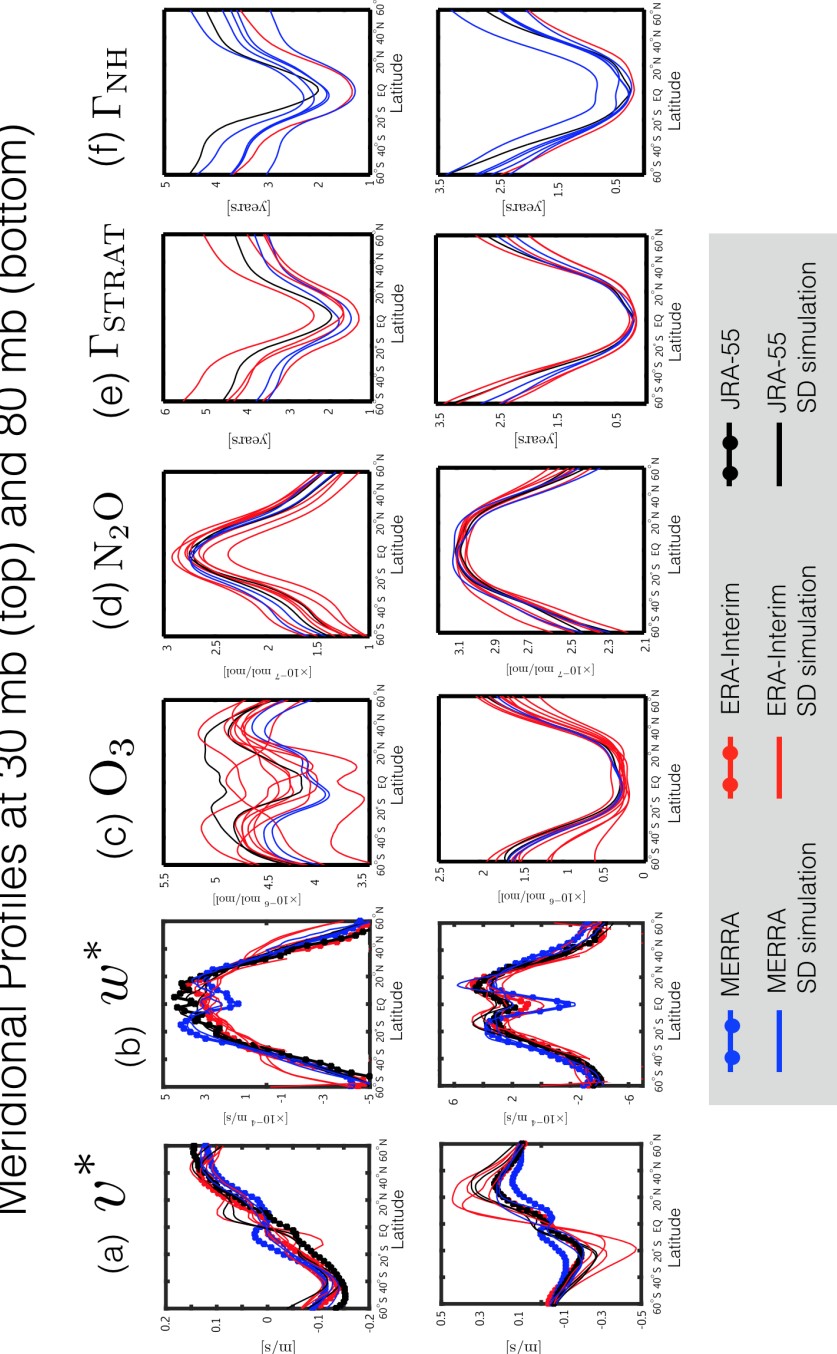

**Figure 2.** Meridional profiles of the 2000-2009 climatological annual mean (a) residual meridional velocity ($v^*$) (b) residual vertical velocity ($w^*$) (c) ozone ($O_3$) (d) nitrous oxide ($N_2O$) (e) stratospheric mean age ($\Gamma_{STRAT}$) and (f) Northern Hemisphere midlatitude mean age ($\Gamma_{NH}$). Profiles are shown for 30 hPa (top) and 80 hPa (bottom). Red/blue/black lines correspond to SD simulations constrained with ERA-Interim/MERRA/JRA-55 reanalysis fields. For the cases of $v^*$ and $w^*$ red/blue/black dotted lines correspond to the S-RIP TEM velocities derived from ERA-Interim/MERRA/JRA-55 meteorological fields.

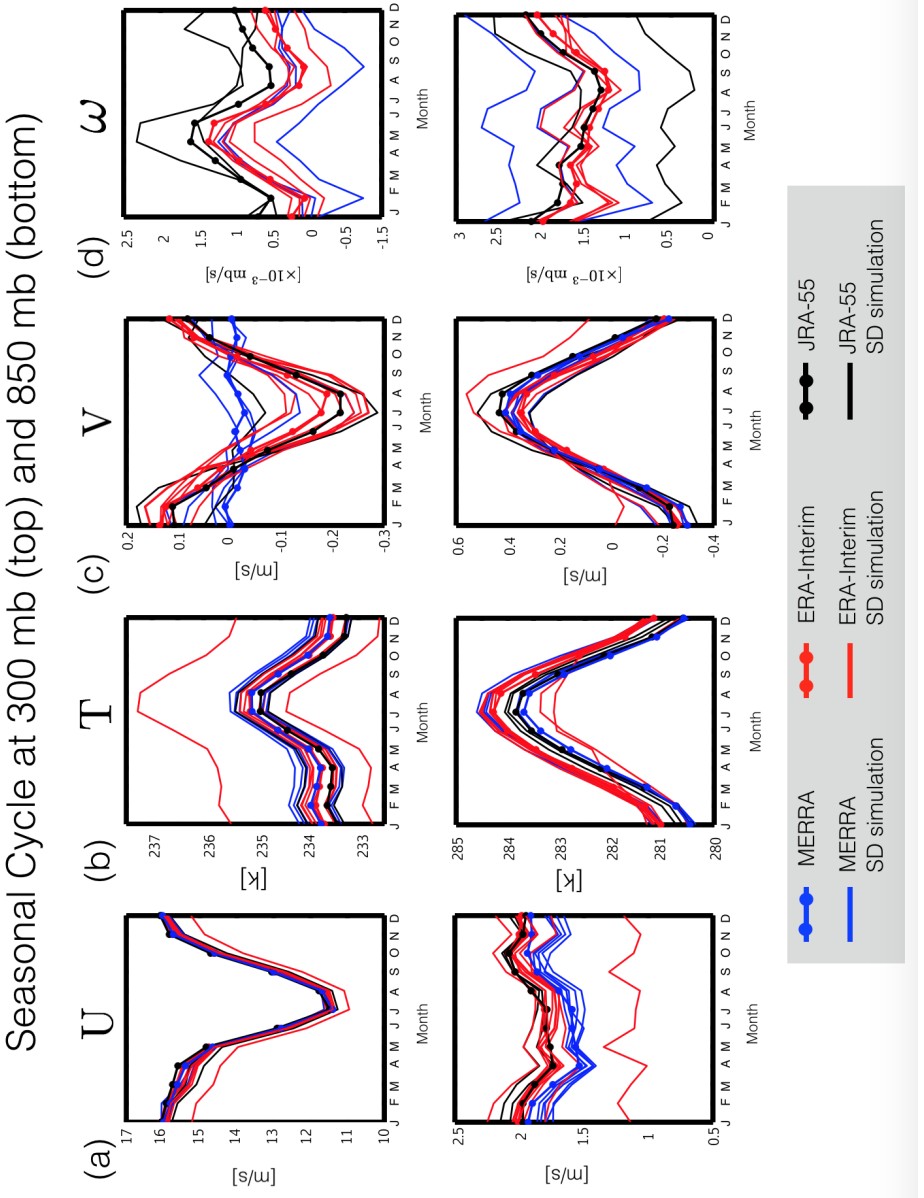

**Figure 3.** 2000-2009 climatological mean seasonal cycle of the (a) zonal mean zonal wind (U) (b) zonal mean temperature (T) (c) zonal mean meridional wind and (d) zonal mean pressure velocity ($\omega$) at 300 hPa (top panels) and 850 hPa (bottom panels). Seasonal cycles have been averaged over latitudes spanning 60°S and 60°N (30°S and 30°N for $\omega$). Red/blue/black solid lines correspond to SD simulations constrained with ERA-Interim/MERRA/JRA-55 reanalysis fields. Red/blue/black dotted lines correspond to the raw ERA-Interim/MERRA/JRA-55 meteorological fields. As in Figure 1 note that MERRA assimilated fields are shown, for which $\omega$ is not available.

# Seasonal Cycle Phase ($\tau_{\min/\max}$) and Amplitude (SCA) at 300 mb (left) and 850 mb (right)

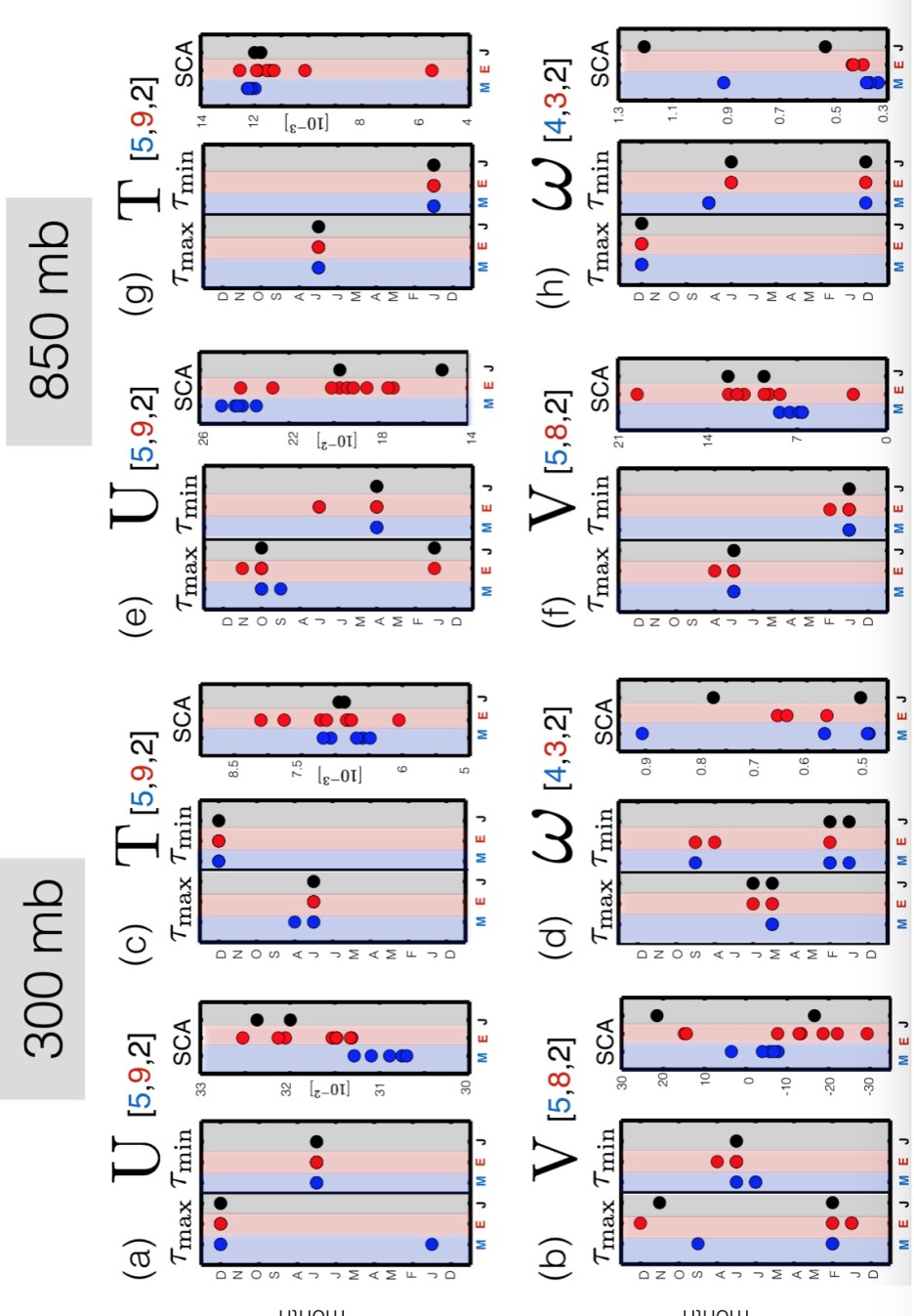

**Figure 4.** The seasonal cycle phase, represented in terms of $\tau_{\min}$ and $\tau_{\max}$, and seasonal cycle amplitude (SCA) of the zonal mean zonal winds (a,e), zonal mean temperatures (c,g), zonal mean meridional winds (b,f) and zonal mean pressure velocities (d,h). Each dot represents an individual model simulation. The seasonal cycle amplitude has been normalized by the climatological mean annually averaged value for each variable shown. Red/blue/black circles show the spread among the SD simulations constrained with ERA-I/MERRA/JRA-55 reanalysis fields. Right and left panels correspond to evaluations at 300 hPa and 850 hPa, respectively. Note that the number of ensembles members per ensemble and for each variable are shown in the title within each panel. In addition, note that in several panels only a few dots are visible. This reflects the overall good consistency in seasonal cycle phase among the simulations.

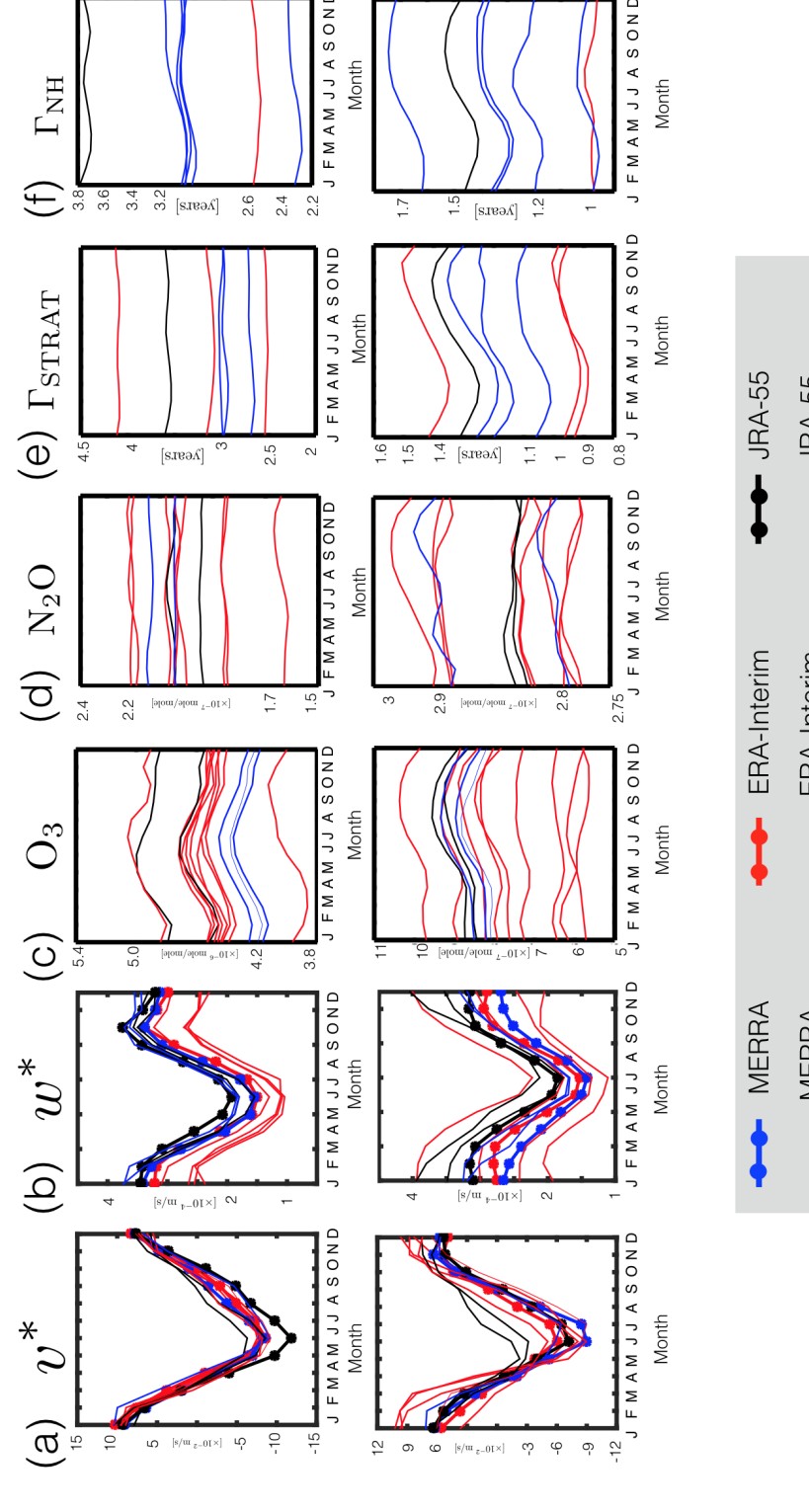

**Figure 5.** Same as Figure 3, but now for the stratospheric dynamical and transport diagnostics. Shown are the 2000-2009 climatological mean seasonal cycles of the (a) residual mean meridional velocity ($v^*$) (b) residual mean vertical velocity ($w^*$) (c) ozone ($O_3$) (d) nitrous oxide ($N_2O$) (e) stratospheric mean age ($\Gamma_{\text{STRAT}}$) and (f) NH midlatitude mean age ($\Gamma_{\text{NH}}$). The seasonal cycle is shown at 30 hPa (top panels) and 80 hPa (bottom panels) and all latitudinal averages have been performed over 60°S and 60°N (30°S and 30°N for $w^*$). For the cases of $v^*$ and $w^*$ red/blue/black dotted lines correspond to the S-RIP TEM velocities derived ERA-Interim/MERRA/JRA-55 meteorological fields..

# Seasonal Cycle Phase ($\tau_{min/max}$) and Amplitude (SCA) at 30 mb (top) and 80 mb (bottom)

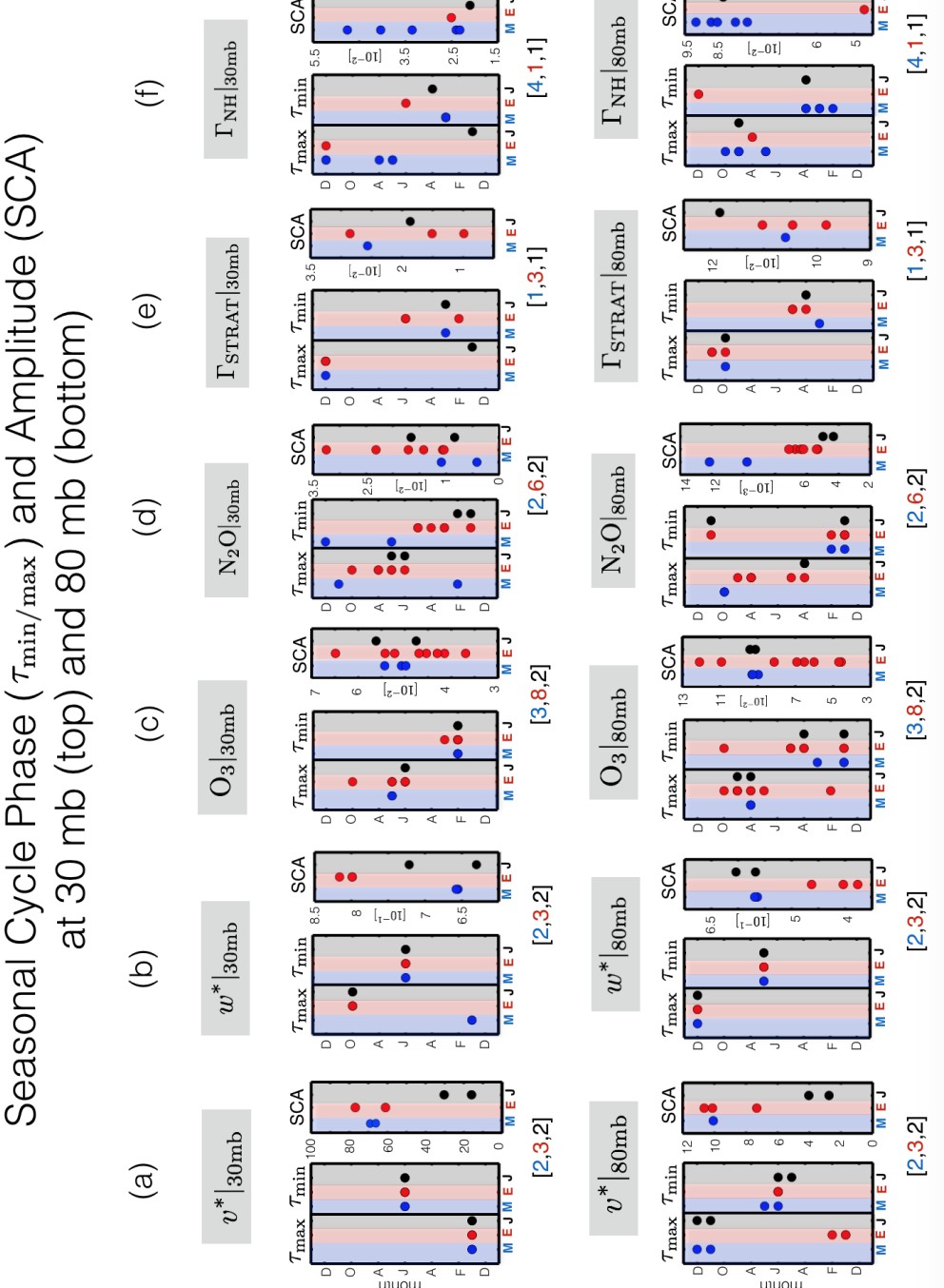

**Figure 6.** Same as Figure 4, but now for the stratospheric dynamical and transport diagnostics.

# Timeseries at 300 mb (left) and 850 mb (right)

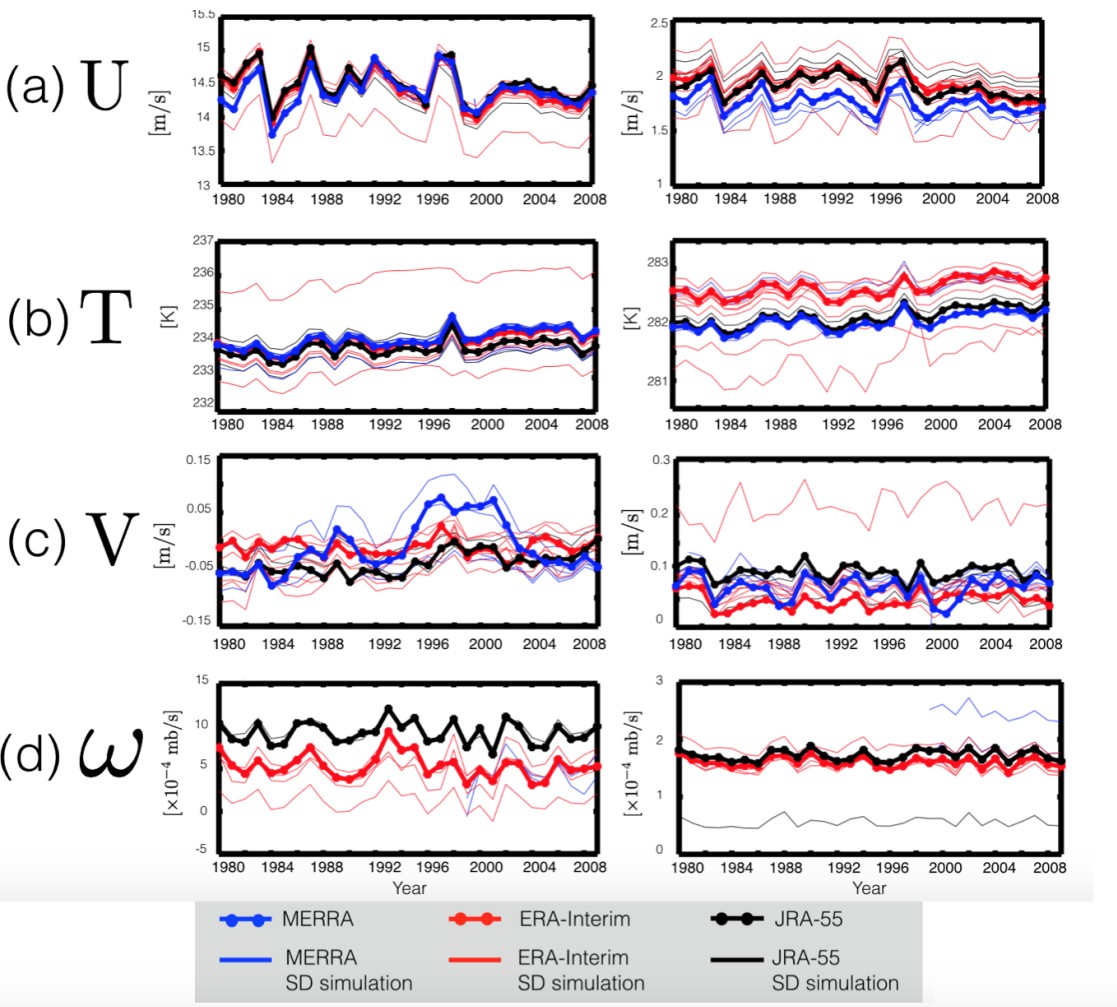

**Figure 7.** Timeseries of the zonal mean zonal winds (a), zonal mean temperatures (b), zonal mean meridional winds (c) and zonal mean vertical velocities (d) at 300 hPa (left panels) and 850 hPa (right panels). Red/blue/black solid lines correspond to SD simulations constrained with ERA-Interim/MERRA/JRA-55 reanalysis fields. Red/blue/black dotted lines correspond to the raw ERA-Interim/MERRA/JRA-55 meteorological fields. MERRA assimilated fields are shown for all variables, except for $\omega$, for which output was not available.

# Timeseries at 30 mb (left) and 80 mb (right)

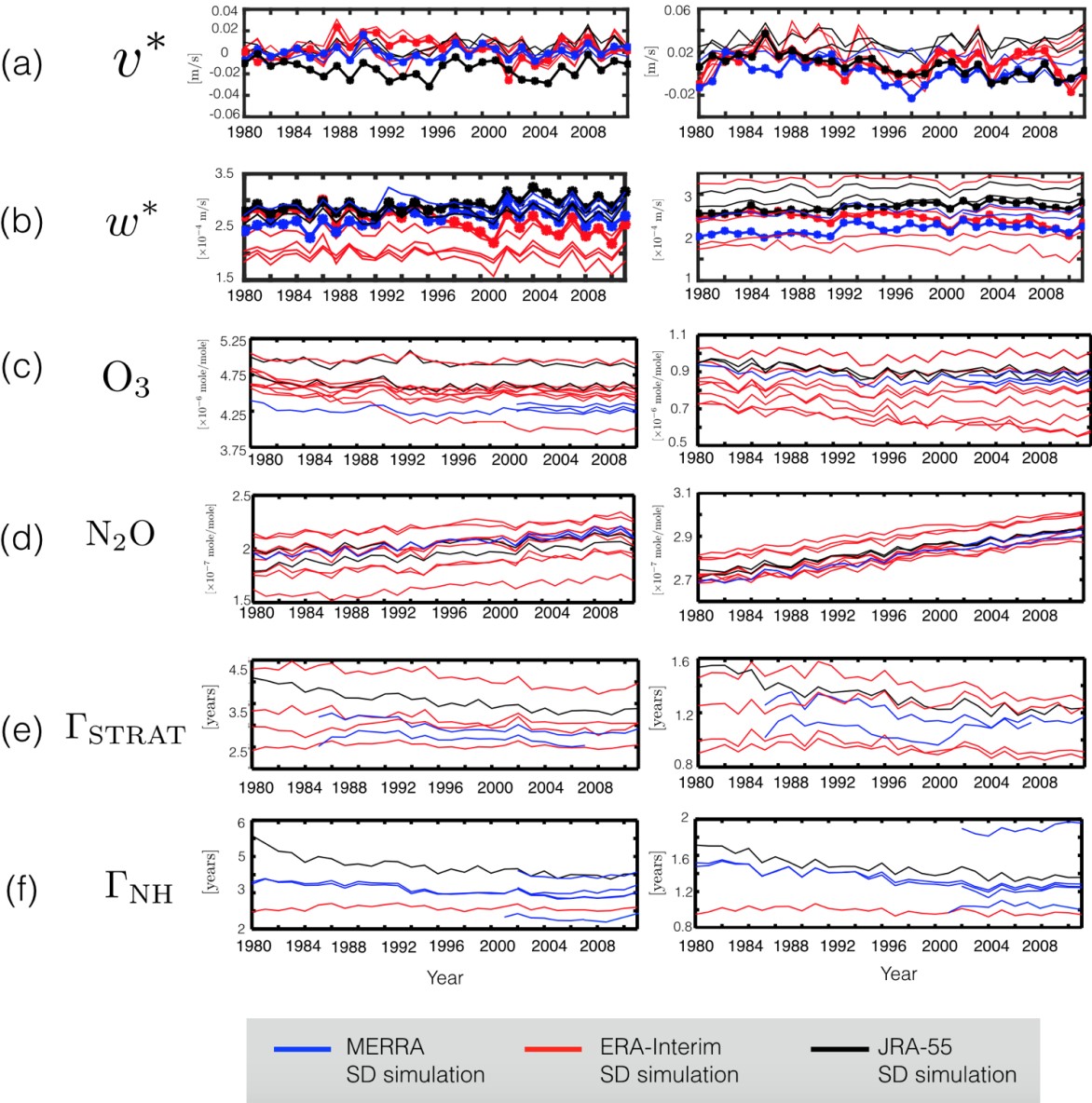

**Figure 8.** Same as Figure 7, but for the stratospheric dynamical and transport fields, evaluated at 30 hPa (left) and 80 hPa (right). For the cases of $v^*$ and $w^*$ red/blue/black dotted lines correspond to the S-RIP TEM velocities derived from ERA-Interim/MERRA/JRA-55 meteorological fields.

Trends of Tropospheric (top) and Stratospheric (bottom) measures at 300 mb/30 mb (left) and 850 mb /80 mb (right)

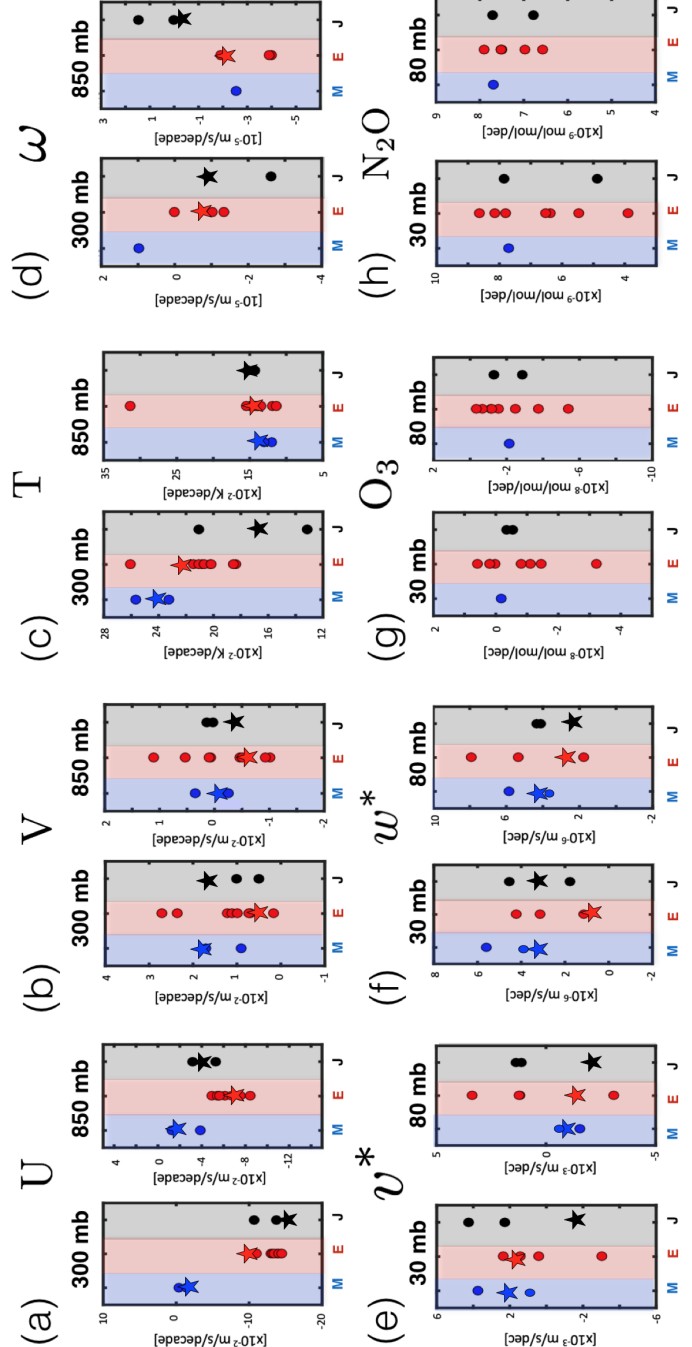

**Figure 9.** Top: Trends in the zonal mean zonal winds (a), meridional winds (b), temperatures (c) and pressure velocities (d), evaluated as simple linear fits of the timeseries shown in Figure 7 over years 1980–2009. Bottom: Same, but for the (e) residual mean meridional velocity (f) residual mean vertical velocity (g) ozone and (h) nitrous oxide. Red/blue/black circles denote the SD simulations, whereas the stars refer to the trends inferred directly from the S-RIP TEM velocities derived from ERA-I/MERRA/JRA-55 reanalysis fields.

# Root Mean Square (RMS) Error over 60°S-60°N

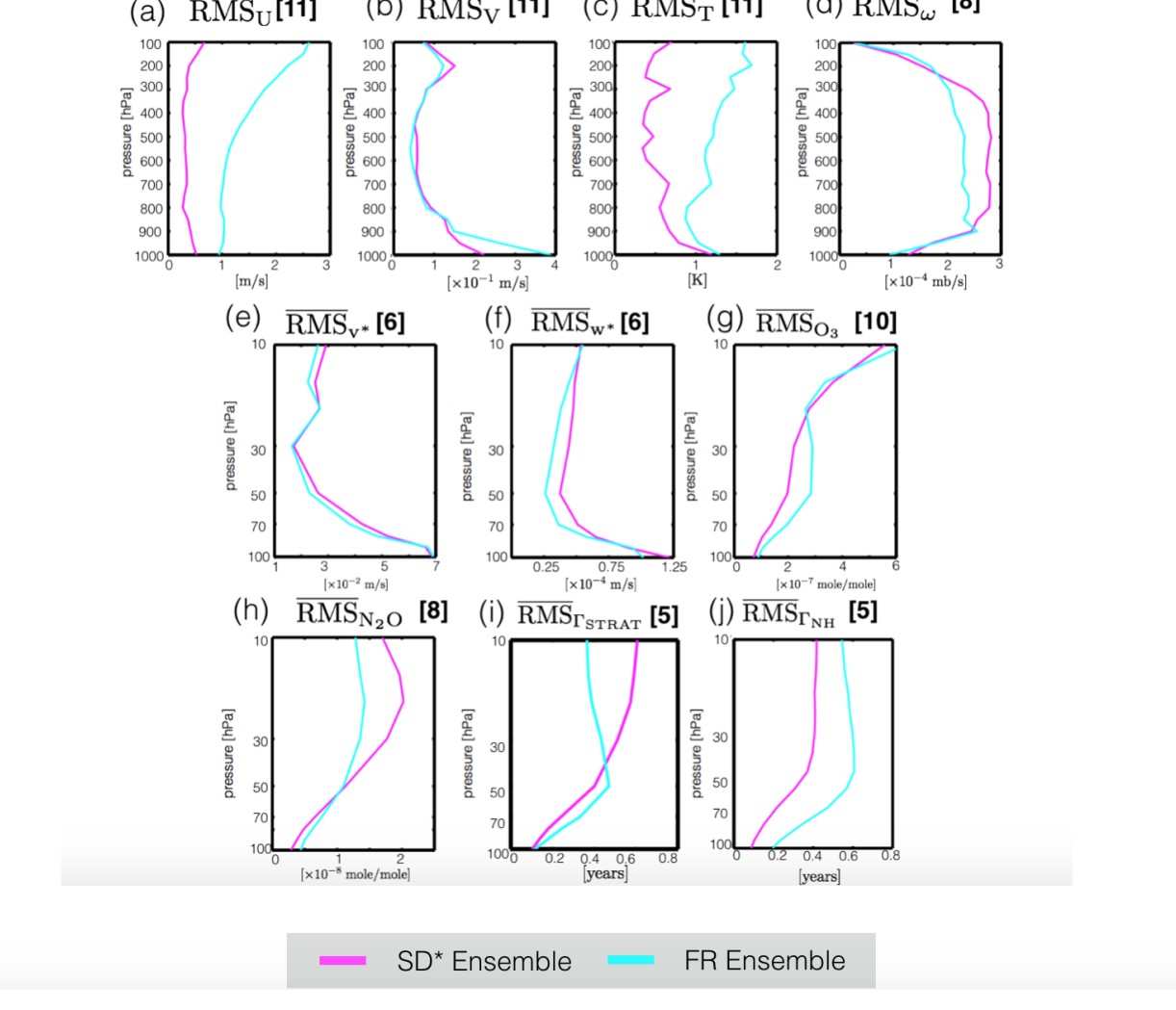

**Figure 10.** The Root Mean Square (RMS) spread of the (a) zonal mean zonal winds (b) zonal mean meridional winds (c) zonal mean temperatures (d) zonal mean pressure velocities (e) residual mean horizontal velocity (f) residual mean vertical velocity (g) ozone (h) nitrous oxide (i) stratospheric mean age and (j) NH midlatitude mean age. The pink and cyan lines correspond to the ensemble mean RMS spread evaluated over the SD* and FR ensembles, where the asterisk denotes the subset of SD simulations for which there was a corresponding free-running simulation (last column in Table 1).

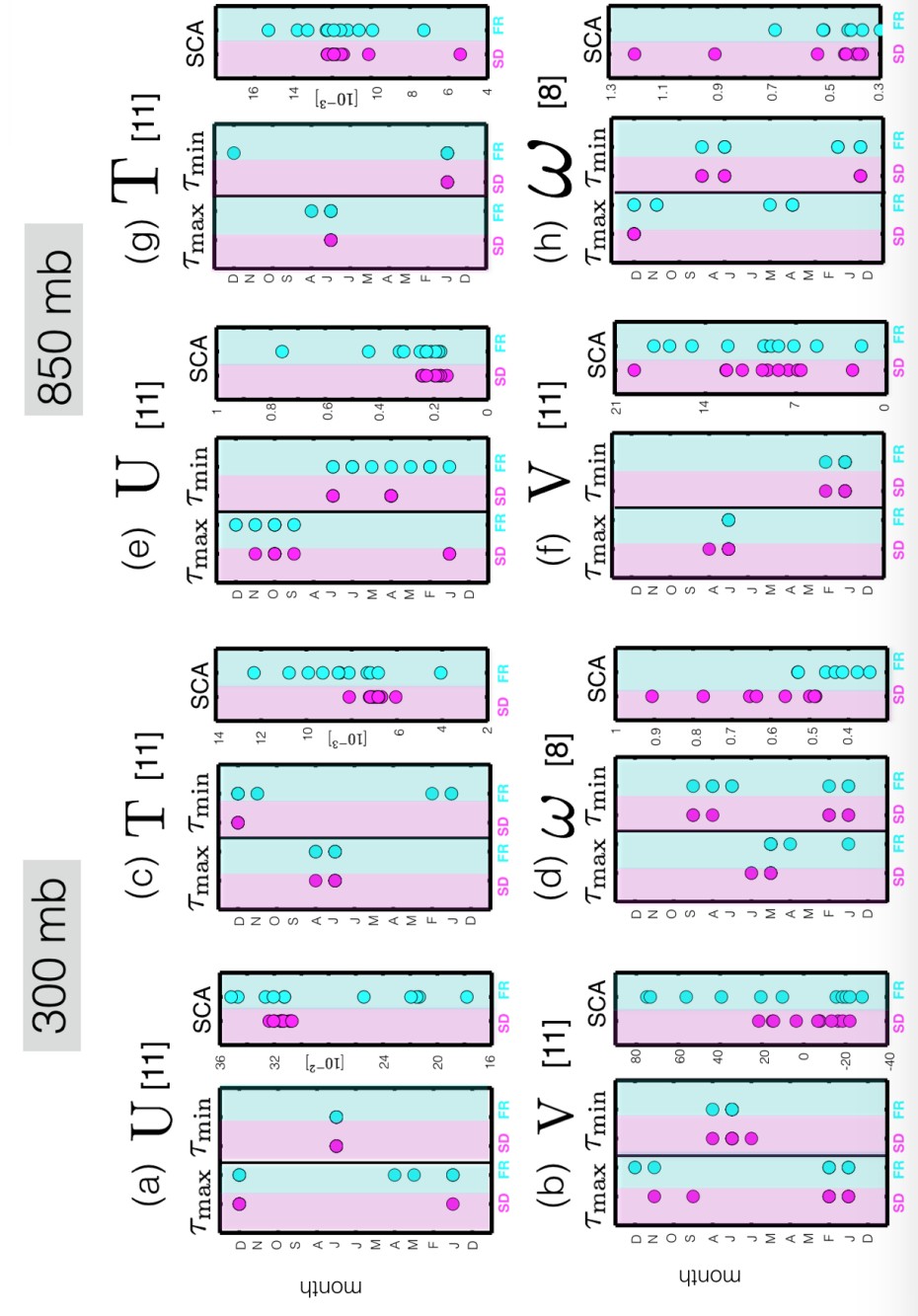

**Figure 11.** Same as Figure 4, but now evaluated for the SD* and FR ensembles in pink and cyan, respectively. Note that both ensembles contain the same number of ensemble members per variable, as shown in the title within each panel.

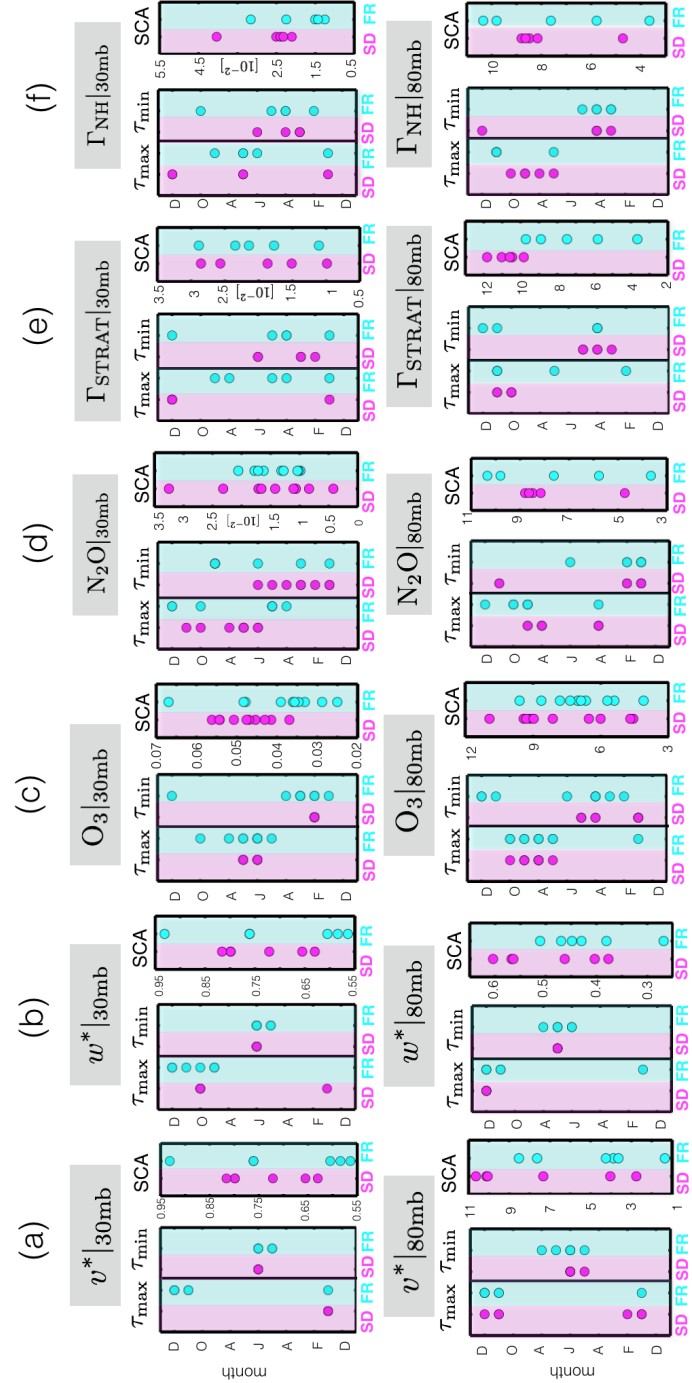

**Figure 12.** Same as Figure 11, but now evaluated for the stratospheric dynamical and transport variables, shown here at 30 hPa (top) and 80 hPa (bottom).

Trends of Tropospheric (top) and Stratospheric (bottom) measures at 300 mb/30 mb (left) and 850 mb /80 mb (right)

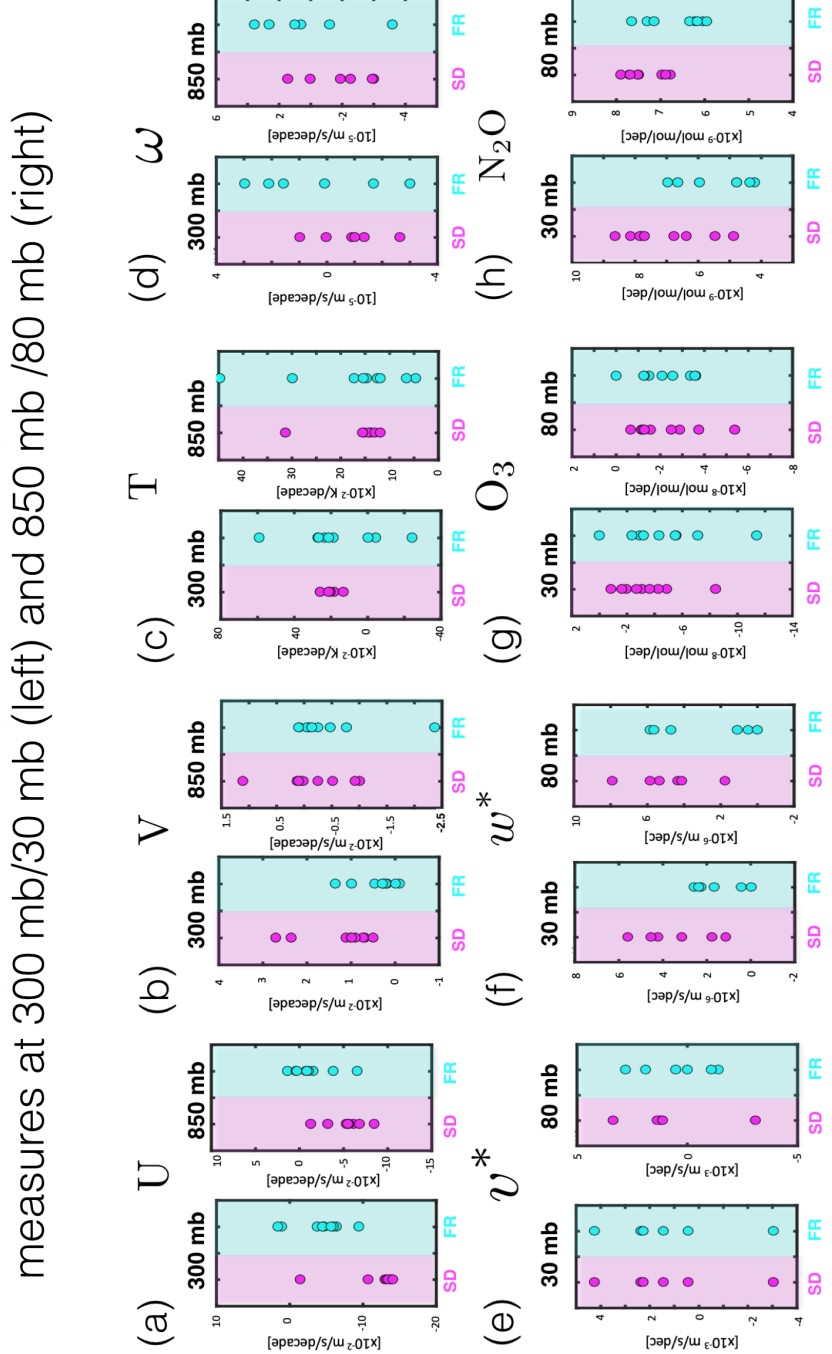

**Figure 13.** Same as Figure 9, but now evaluated for the SD* and FR ensembles in pink and cyan.

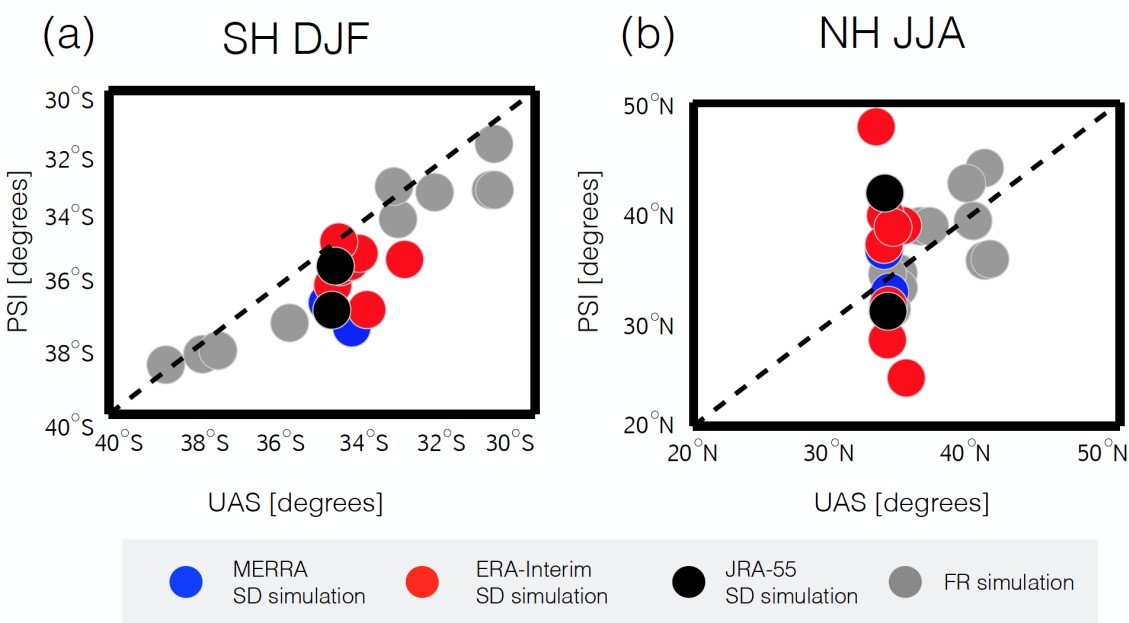

**Figure 14.** Correlations of PSI and UAS, two independent measures of the Hadley Cell edge calculated from the meridional winds and zonal winds, respectively, evaluated over years 1980-2009 for both the SD* and FR* ensembles. Correlations are shown for SH DJF (left) and NH JJA (right), with red/blue/black circles corresponding to ERA-I/MERRA/JRA-55 simulations while grey circles denote free-running simulations.

| REF-C1SD Simulation Name | CTM or Nudged | Source of Met. Fields | Constrained Fields | Nudging Timescales ($\tau$) |
|---|---|---|---|---|
| GEOS C1SD* | Nudged | MERRA† | U, V, T, PS | N/A |
| GMI CTM | CTM | MERRA | U, V | N/A |
| WACCM-5(50)hr* C1SD | Nudged | MERRA† | U, V, T, PS, TAUX, TAUY, SHFLX, LHFLX | $5(50)\ \mathrm{hrs}^{-1}$ |
| CAM C1SD* | Nudged | MERRA | U, V, T, PS, TAUX, TAUY, SHFLX, LHFLX | $50\ \mathrm{hrs}^{-1}$ |
| EMAC-L47(L90)* C1SD | Nudged | ERA-Interim | D, $\zeta$, T, ln(PS) | $6\text{-}48\ \mathrm{hrs}^{-1}$ |
| MRI C1SD* | Nudged | JRA-55 | U, V, T | $24\ \mathrm{hrs}^{-1}$ |
| CMAM C1SD* | Nudged | ERA-Interim | D, $\zeta$, T | $24\ \mathrm{hrs}^{-1}$ |
| NIES C1SD* | Nudged | JRA-55 | U, V, T | $24\ \mathrm{hrs}^{-1}$ |
| CHASER C1SD | Nudged | ERA-Interim | U, V, T | $0.8\text{-}8\ \mathrm{days}^{-1}$ |
| MOCAGE CTM | CTM | ERA-Interim | U, V, T and Q | N/A |
| CNRM C1SD* | Nudged | ERA-Interim | U, V, T and Q | $5\ \mathrm{hrs}^{-1}$ |
| UMUKCA C1SD* | Nudged | ERA-Interim | U, V, $\theta$ | $6\ \mathrm{hrs}^{-1}$ |
| IPSL C1SD* | Nudged | ERA-Interim | U, V, T | $3\ \mathrm{hrs}^{-1}$ |
| TOMCAT CTM | CTM | ERA-Interim | D, $\zeta$, T, PS | N/A |
| GFDL C1SD** | Nudged | NCEP | U, V | $6\text{-}600\ \mathrm{hrs}^{-1}$ |
| HadGEM C1SD** | Nudged | ERA-Interim | U, V, T | $1\ \mathrm{day}^{-1}$ |

**Table 1.** List of the REF-C1SD model simulations that were conducted in support of CCMI-Phase 1. Single asterisks in Column 1 indicate that a corresponding free-running REF-C1 simulation was also performed using the same underlying model code. Columns 2 and 3 indicate whether REF-C1SD was produced using an offline CTM or online nudging, and the source of the constraining analysis fields, respectively. Columns 4 and 5 list the fields that were constrained in the REF-C1SD simulations, with D and $\zeta$ denoting the divergence and vorticity fields, respectively, as well as the timescales over which nudging was performed. For the WACCM and CAM simulations, additional nudging was performed with respect to TAUX, TAUY, SHFLX, LHFLX (surface stress and latent and sensible heat fluxes). Note that Column 5 only broadly summarizes the nudging timescales applied in the REF-C1SD simulations for sake of brevity. We refer the reader to the Appendix for more information about cases where $\tau$ exhibited a functional dependence (on pressure, for example).

†Note that the GEOS-Replay and WACCM C1SD simulations are constrained with MERRA *analysis* fields. For more on the difference between the analysis and assimilated fields we refer the reader to Orbe et al. (2017b).

**The output for these simulations was not available for analysis.

| REF-C1SD Simulation or Reanalysis Product | U | V | T | $\omega$ | $v^*$ | $w^*$ | $O_3$ | $N_2O$ | $\Gamma_{STRAT}$ | $\Gamma_{NH}$ |
|---|---|---|---|---|---|---|---|---|---|---|
| GEOS C1SD | ✓ | ✓ | ✓ | ✓ | ✗ | ✗ | ✗ | ✗ | ✓ | ✓ |
| GMI CTM | ✓ | ✓ | ✓ | ✗ | ✗ | ✗ | ✗ | ✗ | ✗ | ✓ |
| WACCM-5(50)hr-C1SD* | ✓ | ✓ | ✓ | ✓ | ✓ | ✓ | ✓ | ✓ | ✓ | ✓ |
| CAM C1SD | ✓ | ✓ | ✓ | ✓ | ✗ | ✗ | ✗ | ✗ | ✗ | ✓ |
| EMAC-L47(L90) C1SD | ✓ | ✓ | ✓ | ✓ | ✓ | ✓ | ✓ | ✓ | ✓ | ✗ |
| MRI C1SD | ✓ | ✓ | ✓ | ✓ | ✓ | ✓ | ✓ | ✓ | ✓ | ✓ |
| CMAM C1SD | ✓ | ✓ | ✓ | ✓ | ✓ | ✓ | ✓ | ✓ | ✓ | ✓ |
| NIES C1SD | ✓ | ✓ | ✓ | ✓ | ✓ | ✓ | ✓ | ✓ | ○ | ○ |
| CHASER C1SD | ✓ | ✓ | ✓ | ✗ | ✗ | ✗ | ✓ | ✓ | ✗ | ✗ |
| MOCAGE CTM | ✓ | ✗ | ✓ | ✗ | ✗ | ✗ | ✓ | ✗ | ○ | ○ |
| CNRM C1SD | ✓ | ✓ | ✓ | ✗ | ✗ | ✗ | ✓ | ✗ | ✗ | ✗ |
| UMUKCA C1SD | ✓ | ✓ | ✓ | ✗ | ✗ | ✗ | ✓ | ✓ | ✗ | ✗ |
| IPSL C1SD | ✓ | ✓ | ✓ | ✗ | ✗ | ✗ | ✓ | ✓ | ✗ | ✗ |
| TOMCAT CTM | ✓ | ✓ | ✓ | ✗ | ✓ | ✓ | ✓ | ✓ | ✓ | ✗ |
| GFDL-AM3 C1SD | ✗ | ✗ | ✗ | ✗ | ✗ | ✗ | ✗ | ✗ | ✗ | ✗ |
| HadGEM3 C1SD | ✗ | ✗ | ✗ | ✗ | ✗ | ✗ | ✗ | ✗ | ✗ | ✗ |
| MERRA | ✓ | ✓ | ✓ | ✗ | ✗ | ✗ | ✗ | ✗ | ✗ | ✗ |
| ERA-Interim | ✓ | ✓ | ✓ | ✓ | ✗ | ✗ | ✗ | ✗ | ✗ | ✗ |
| JRA-55 | ✓ | ✓ | ✓ | ✓ | ✗ | ✗ | ✗ | ✗ | ✗ | ✗ |

**Table 2.** List of the model simulations for which the dynamical fields (U, V, T, $\omega$, $v^*$ and $w^*$) and both chemical and idealized tracers ($O_3$, $N_2O$, $\Gamma_{NH}$ and $\Gamma_{STRAT}$) were available as (monthly mean) output. Checkmarks (crosses) denote fields that were (were not) output in simulations. Note the same output was also available for all corresponding free-running runs in those cases in which both REF-C1 and REF-C1SD simulations were provided (see Column 1, Table 1). Open circles denote tracers that were integrated in simulations, but not implemented correctly. *Output for WACCM-5hr was available only for years 2000-2009 for several variables.

| Source of SD Differences | Description | Examples |
|---|---|---|
| Reanalysis Fields | Differences in the fields from the different reanalysis products. | SCA of $V_{300hPa}$ $\omega_{300hPa}$ |
| Nudging Techniques | Differences in variables, timescales, and spatial domain in nudging. | $v^*$, $w^*$, V, $\omega$, $\overline{\text{corr}}_{T,850hPa}$ |
| Underlying Free-Running Models | Differences due to biases in the underlying climate model. | T at 850 hPa |

**Table 3.** Above we identify three distinct sources of differences among the SD simulations. We distinguish between differences associated with the use of different reanalysis products (row 1) versus differences associated with how the large-scale flow fields are specified (row 2) as well as underlying free-running model biases (row 3). All sources are present in the REF-C1SD ensemble considered in this study.

| Variable ($\chi$) | $\overline{\mathrm{corr}}_{\chi,\mathrm{SD}}$ | $\overline{\mathrm{corr}}_{\chi,\mathrm{MERRA}}$ | $\overline{\mathrm{corr}}_{\chi,\mathrm{ERAI}}$ | $\overline{\mathrm{corr}}_{\chi,\mathrm{JRA55}}$ | ‖ | $\overline{\mathrm{corr}}_{\chi,\mathrm{SD}^*}$ | $\overline{\mathrm{corr}}_{\chi,\mathrm{FR}}$ |
|---|---|---|---|---|---|---|---|
| $U\|_{300}$ | **0.94** | 0.92 | 0.99 | 0.98 | ‖ | **0.97** | **0.89** |
| $U\|_{850}$ | **0.93** | 0.95 | 0.93 | 0.99 | ‖ | **0.93** | **0.82** |
| $V\|_{850}$ | **0.73** | 0.60 | 0.78 | 0.98 | ‖ | **0.71** | **0.59** |
| $V\|_{300}$ | **0.78** | 0.90 | 0.92 | 0.91 | ‖ | **0.75** | **0.45** |
| $T\|_{300}$ | **0.95** | 0.97 | 0.94 | 0.99 | ‖ | **0.94** | **0.69** |
| $T\|_{850}$ | **0.83** | 0.80 | 0.70 | 0.99 | ‖ | **0.85** | **0.71** |
| $\omega\|_{300}$ | **0.88** | 0.89 | 0.99 | 0.79 | ‖ | **0.86** | **0.35** |
| $\omega\|_{850}$ | **0.84** | 0.80 | 0.95 | 0.71 | ‖ | **0.84** | **0.48** |
| $O_3\|_{30}$ | **0.73** | 0.74 | 0.80 | 0.77 | ‖ | **0.74** | **0.60** |
| $O_3\|_{80}$ | **0.90** | 0.96 | 0.88 | 0.97 | ‖ | **0.86** | **0.61** |
| $N_2O\|_{30}$ | **0.95** | 0.98 | 0.97 | 0.99 | ‖ | **0.97** | **0.95** |
| $N_2O\|_{80}$ | **0.99** | 0.99 | 0.99 | 0.99 | ‖ | **0.97** | **0.98** |
| $v^*\|_{30}$ | **0.87** | 0.84 | 0.92 | 0.85 | ‖ | **0.87** | **0.34** |
| $v^*\|_{80}$ | **0.78** | 0.97 | 0.96 | 0.87 | ‖ | **0.75** | **0.33** |
| $w^*\|_{30}$ | **0.93** | 0.98 | 0.98 | 0.97 | ‖ | **0.91** | **0.50** |
| $w^*\|_{80}$ | **0.92** | 0.92 | 0.95 | 0.98 | ‖ | **0.91** | **0.58** |
| $\Gamma_{\mathrm{STRAT}}\|_{30}$ | **0.90** | 0.89 | 0.98 | 1 | ‖ | **0.81** | **0.85** |
| $\Gamma_{\mathrm{STRAT}}\|_{80}$ | **0.81** | 0.83 | 0.84 | 1 | ‖ | **0.84** | **0.87** |
| $\Gamma_{\mathrm{NH}}\|_{30}$ | **0.91** | 0.70 | 0.98 | 1 | ‖ | **0.91** | **0.68** |
| $\Gamma_{\mathrm{NH}}\|_{80}$ | **0.87** | 0.74 | 0.98 | 1 | ‖ | **0.89** | **0.60** |

**Table 4.** Correlation coefficients for each variable $\chi$ among the SD, MERRA, ERA-Interim, JRA-55, SD$^*$ and FR ensembles. Note that the SD$^*$ ensemble consists only of those SD simulations for which there was also a corresponding FR simulation. Specifically, for the SD ensemble $\mathrm{corr}(i)_{\chi,\mathrm{SD}}$ corresponds to the correlation coefficient between the $i^{th}$ member timeseries of variable $\chi$ and the SD ensemble mean timeseries. Averaging over all SD ensemble members we have $\overline{\mathrm{corr}}_{\chi,\mathrm{SD}} \equiv \frac{1}{\mathrm{N}} \sum_{\mathrm{i}=1}^{\mathrm{N}} \mathrm{corr}(i)_{\chi,\mathrm{SD}}$. For all variables correlations were evaluated over years 1980-2009.