# Peer review of "Description and Evaluation of the Specified-Dynamics Experiment in the Chemistry-Climate Model Initiative (CCMI)"

_Atmospheric Chemistry and Physics, 2019_

## Referee Comment (RC1) · Anonymous Referee #1 · 11 Sep 2019

The study by Orbe et al evaluates the performance of the so-called "specified dynamics" simulations performed within the CCMI1 project in terms of tropospheric and stratospheric large-scale circulation as well as stratospheric transport tracers. Further, the results are contrasted with the free-running simulations with the same set of models. The study shows that while zonal winds and temperature climatologies are generally well constrained by "nudging", the meridional and vertical component of the circulation are not. While all variables are constrained in terms of their interannual variability and (less so) seasonal cycle phasing, the seasonal cycle amplitude is less constrained. Another important result is that the SD simulations are no longer dynamically consistent, which can be expected due to the addition of the non-physical nudging term. Overall, the study is an important contribution to the CCMI evaluation papers, and more so an overdue clarification on the effects of nudging in CCMs. Specifically, it clarifies that "nudging" of the large-scale flow does not necessarily lead to a reduction in the spread even in the large-scale flow itself, and therefore even less so in transport and tracers. The paper can be improved in terms of the presentation and writing, but will be an important contribution after thoroughly revising the text. Furthermore, I have a few suggestions that could emphasize the points of the paper even more.

General comments:

1. Trends: There is no mentioning of long-term trends in the paper. It is found that interannual variability in the SD simulations is rather consistent, and thus it is concluded correctly that the SD simulations are most justified to be used for studies of interannual variability (page 17, lines 6ff). I strongly recommend to add a word of caution here that the agreement of interannual variability very likely will not transfer to long-term trends. Even more so, I suggest to the authors to add a small section that specifically analyses the trends; while this might seem like opening another "can of worms", in my opinion this could be done in a rather short way: If you just calculate the trends of the time-series presented in 7 and 8 and present them in one additional Figure, you clarify how well the trend is constrained (or not). Of course comparison to the FR simulations (and underlying reanalysis were appropriate) would be very valuable here as well. On a similar note, as stated by the authors the time-series were not de-trended before the interannual corrections were calculated, which might obscure the rating of the interannual variability. I recommend to detrend the timeseries for the correlations of interannual variability, and regard the trends separately as suggested above. As long-term trends are the subject of many studies (and you even mention "past trends" in the 2nd sentence of the abstract), and often the naive idea that nudged simulations should get the observed trend right is still around, I think this would be a very important addition to the paper.

2. Reanalysis data of TEM circulation: Is there a particular reason why the analysis of v* and w* is not contrasted to the respective data from the reanalysis? While I understand that calculating those diagnostics is a large effort, they are available from the "SRIP" dataset (see https://www.earth-syst-sci-data.net/10/1925/2018/ and https://catalogue.ceda.ac.uk/uuid/b241a7f536a244749662360bd7839312). If you choose not to include the data, please give a short explanations for the reasons in the paper.

Specific comment:

Abstract: The abstract emphasizes the differences in the SD simulations strongly, but does not mention that the interannual variability is indeed constrained in the simulations. While I agree that the text should clearly state the "warning" to users of SD simulations, it would be fair to also mention the positive outcomes of this study. On the other hand, I think it would be good to mention the results on the dynamical inconsistency in the Abstract, as I think this is a major result to keep in mind when working with SD simulations: the dynamics are not only not well constrained, but they are actually internally consistent (which makes sense as an non-physical term is added to the budgets). This important finding should also be mentioned in the conclusions.

page 3, line 26: I find the way you introduce the possible reasons for the SD differences a little confusing. Why not list all three points first, and then mention that in the following with "implementation differences" you refer to the named two points (and give a reason why you group them - presumably because it is hard to quantify their relative roles?)

page 5, line 3-4: Could the transformation of w to omega contribute to model spread in this variable, as probably the density is approximately calculated using the given zonal mean monthly mean temperatures? What is the reasoning behind not using w?

page 5, line 22: why is the interpolation for "tropospheric variables" performed? Because the CCMI model levels are too coarse in the troposphere?

page 6, line 12: before mentioning ensemble member, define that the ensemble here is

the multi-model suit of SD experiments (earlier, it was mentioned that there are multiple ensembles of the REF-C1 simulations in the conventional meaning of "ensemble", so this is somewhat confusing).

page 6, line 24, and entire section 3: In general, I find your way of using "e.g." when listing models a little confusing - this implies that you only list examples, when indeed you do list all the models with certain properties. So please just remove the "e.g." (see also page 7, line 23 and 26). Similarly, when mentioning that "few" models have a certain property, please specify the number. For example in Section 3.3 you say "most" models use HadISST, while "some" use other forcing. While I agree you do not need to list all details here, please be a bit more specific, otherwise this information is more confusing that helpful.

page 9, line 21: the listed differences occur quite close to the equator, where the variable changes it sign, does it? As those result from small shifts in the distribution, it might not be entirely fair to list those as range or difference - rather refer for example to difference in maximum values?

page 9, line 30ff: Good to see you applied the recalculation to check on the possible impact of this issue. According to the Figure in the supplement, the magnitude of differences appears to be larger to me in the recalculated w* values? Therefore I would recommend to reformulate that statement, in that the differences are at least as high as the ones shown in Fig. 2 (or in other words, that the differences cannot be explained by differences in the calculation method of w*, as the recalculation even emphasizes the differences.)

page 10, line 1-2: Because they are clustered by reanalysis product, correct? As you do not present the values from the reanalysis this is indirectly inferred.

page 10, line 5: I find the references to Table 3 mentioned here (and elsewhere) not very helpful. In particular in this paragraph, there is no explanation of why the respective potential reasons are thought to explain the differences. In general, I think Table

3 could be removed, and rather the potential reasons should be mentioned/discussed where appropriate.

page 11, line 1: Isnt the spread in U850 almost as large?

page 11, line 3ff: I think here you mix up the phase and the amplitude: the spread in V300 tau_max is present for all three RA (largest for MERRA, but still 3-4 months for the other RA). If you refer to the amplitude rather than phase here, this would make more sense (then the text needs to be rephrased).

Fig. 4: Please specify more clearly what the individual "dots" represent for the phase: are those the individual models? (and only few are seen in the phase plot because they lay on top of each other?). In the caption it says "show the spread...", which does not clarify what is shown.

page 11, line 17: Could the effect of small annual means amplify the differences? If one model has zero mean, its annual cycle would be infinity...?

page 12, line 16: Do you really mean "positive correlations" here, or "high"? Negative correlations would be very surprising.

page 13, line 9: could you mention the value for N here to give the reader a feeling of the ensemble size (without having to go to Table 1 and count).

Fig. 9: I really like this Figure and think it provides valuable information!

page 14, top: I also like the Figures 4 and 5 of the supplement (but agree it is too much to show them in the paper). Some additional interesting features are to be identified, e.g. for omega, it turns out that the RMS at higher latitudes is stronger for FR than for SD. Also for w*, this Fig highlights that the RMS appears to be generally higher in the SD simulations, a result that I'm not surprised by. Mentioning of those interesting features could be worthwhile.

page 14, line 12: I'd summarize the findings in that SCA is (much) better constrained

for T und U for the SD runs, and similar for V, but the spread in the SCA is even larger in the SD runs for omega.

page 14, line 16: "poorly (equally)" - confusing formulation, please clarify

page 15, top: Again, one interesting result that could be mentioned here is that the correlations for w* are lower in the SD simulations in the tropics compared to FR, a result that is not as clear from Fig. 11 (where the spread is similar or lower in SD).

page 17, line 35: Do you need to make the assumption that advection scheme biases are small? I think you can rate it under "underlying free-running model" biases. So even if the circulation would be constrained perfectly (which it is not, see TEM diagnostics), one would expect the advection schemes to induced differences, but those should be smaller than in the FR simulation - but they are about as high (see Fig. 9), i.e. spread in the circulation itself must contribute.

page 18, top: I recommend to include somewhere in the paper the important result of dynamical inconsistency. Also, you can refer to Chrysantou et al. for the dynamical inconsistency of the TEM budget (the "downward control" calculation works less well in the SD runs).

Table 1: A few questions to be clarified: Are the models with non-constant nudging timescales those that give a range, and the others have constant nudging timescales? Why include information on models that are not used?

Technical:

Abstract, line 2, and later: the word "online" GCM is not well defined. I would advise not to use it in the Abstract, and if used later on it needs to be defined.

page 3, line 22: I suggest to remove "it is important to note" here, but rather just state that in addition to your paper, there is a study that focuses on the stratospheric circulation (also, the next sentence also starts with "it is important to note")

page 3, line 30: two times "overall" in one sentence.

page 3, bottom: sections should be labels 1,2,3,... as in the text

page 4, line 17: rather introduce the model names than institutions? It is not really scientifically relevant whether the contributions are from the same institution, but whether it is the same model that contributes with different set-ups (and as I understand it, the NASA contributions are two different models). Furthermore, also other models contribute more than one set-up (e.g. EMAC L47 and L90).

page 6, line 30: add "information" after "for more"

page 9, line 11 and general: mbar or hPa? There is a mixture, and as this an European journal you might just want to use hPa.

page 11, line 3: please put for example V300 in brackets after "the latter", otherwise hard to follow for the reader.

page 11, line 9: again, pleas put e.g. meridional winds in brackets after "the former"

page 15, line 5: remove "e.g." (?)

---

## Referee Comment (RC2) · Anonymous Referee #2 · 25 Sep 2019

General Comments

Orbe et al. provide a systematic description of the specified dynamics (SD) model experiments carried out as part of the CCMI exercise. They also evaluate how well the model experiments represent various aspects of the atmospheric state and large scale flow. In some instances, large differences are reported, and are related to differences in the reanalysis products used between the different model experiments, as well as differences in the implementation of the specification of the dynamics, which are themselves further analysed both in terms of inter-model differences in the ways in which the large scale flow was constrained, and biases in the underlying free running

models. Meridional and vertical winds were found to be quite poorly represented in the SD model experiments.

The manuscript is necessarily quite dense and packed with information, but overall I find it to be well structured and logically organised. A high level of detail is presented in an accessible way, caveats are well explained, and suitable interpretation of the results is usually given. I think this manuscript will be a good resource for the CCMI community (and beyond). I recommend it for publication in ACP, with only a small number of minor revisions.

Specific Comments

Language: In general the manuscript is clear and relatively easy to follow, and written in a concise style. In a few places however, some redundant language seems to have crept in to the text. For example on page 9, line 11 "Moving next to the vertical winds" is absolutely not needed in this sentence at all. Similarly, at the beginning of the following paragraph, the sentence could be rewritten without "Next we compare". There are some other examples of this meta-level discussion of what you do first, then what you do next, which for me are simply not necessary. See also page 13, line 12, and page 15, line 3.

Interannual variability: It would be good to be more specific about what you mean by "interannual variability" in the context of this paper. Clearly you are referring to the correlation of annual average quantities from the model experiments for specific years with the corresponding annual averages from the reanalysis products. This is mostly clear by also referring to this as "covariability". But perhaps some readers might have other ideas about what it means for a model to be representing "interannual variability" well, for example how well periodic variations (eg. ENSO) are represented in a statistical sense (eg. by reproducing probability distributions of their frequency and amplitude), rather than their exact timing. You could avoid this potential confusion by being clearer about what you mean by this term.

Page 2, line 11: Perhaps be explicit that by "boundary conditions" you mean the SSTs and SICs. The flow fields could also be considered a kind of boundary condition.

Page 2, line 13: "improves transport". What would constitute an improvement of transport exactly? It's not clear what you really mean here.

Section 3.4: Can you go into a little bit more detail about why the different groups chose such different strategies for their choice of nudged fields, domains, and timescales? Can you synthesise relevant parts of earlier work by the respective modelling groups, or even draw on unrelated literature to provide more detail about how and why these choices are made? I think some more context could be useful here.

Section 5.3: I think the lack of dynamical consistency in the SD simulations is as interesting a result of this study as any of the other results you discuss, and could probably be highlighted more in the abstract and conclusions. I would also like to see some discussion here about the implications of this for tracer transport, especially with respect to the use of SD for tropospheric chemistry simulations (especially involving long-range transport), and inverse modelling studies of greenhouse gases.

Page 17, lines 18 and 20: The use of the word "cases" is confusing here. It's not immediately clear what a "case" is. Is it maybe a specific model run? With some effort the reader can see that you actually mean variables or fields (eg. T850hPa), so perhaps use "fields" here to be more consistent with the immediately preceeding sentence.

Conclusions: For the benefit of readers who like to read the conclusions section before reading the rest of the paper, it would be good to define all acronyms again in this section. Please consider the following acronyms: SD; TEM; CTM; and CCM. CCMI should be clear enough.

---

## Short Comment (SC1) · 29 Sep 2019

Please note that there are two additional published studies that are relevant for the discussions included in this manuscript and should also be mentioned in the main text. Details provided here below:

Page 2, line 20: Reference to Monge-Sanz et al. (2012) should also be added, in this study we built on the one you already cite and provided the comparison of different temporal sampling using the TOMCAT CTM.

Page 17, around lines 24-26: Please add reference, and brief context discussion, to a

relevant published study in which we provided a comparison of stratospheric transport between TOMCAT offline simulations, free running GCM simulations and the corresponding nudged GCM simulations, using both ERA-40 and ERA-Interim reanalyses (Monge-Sanz et al., 2013). The comparison showed remarkably similar results for the offline and the corresponding GCM nudged simulations (see Fig. 9 in Monge-Sanz et al., 2013).

References: Monge-Sanz BM, Chipperfield MP, Untch A, Morcrette J-J, Rap A, Simmons AJ. 2013. On the uses of a new linear scheme for stratospheric methane in global models: water source, transport tracer and radiative forcing. Atmospheric Chemistry and Physics. 13, pp. 9641-9660, https://doi.org/10.5194/acp-13-9641-2013.

Monge-Sanz BM, Chipperfield MP, Dee DP, Simmons AJ, Uppala SM. 2012. Improvements in the stratospheric transport achieved by a CTM with ECMWF (re)analyses: Identifying effects and remaining challenges. Q. J. R. Meteorol. Soc. 139(672), pp. 654-673, https://doi.org/10.1002/qj.1996.

---

## Referee Comment (RC3) · Anonymous Referee #3 · 11 Oct 2019

In this paper the authors provide a broad-brush comparison of a set of "specified-dynamics" (SD) simulations of chemistry climate models (CCMs) with the underlying reanalyses used to drive the SD simulations as well as the models' free-running versions. They concentrate on differences either due to: 1) different underlying "dynamics" (reanalysis product), 2) different nudging methods, 3) model biases of the underlying GCM. They find that in particular meridional and vertical winds show substantial differences, which also affects tracer distributions. These differences come about primarily due to differences in implementation of the SD method and are found to be of comparable magnitude as differences among to underlying GCM (free-running model versions). The most important take-home message is therefore that analyses of output by these

SD simulations should carefully take into consideration the discussed issues due to implementation of the SD method.

This is a very valuable contribution and I presume will be a core reference for those working with output from SD simulations. Given the wide scope, the analyses necessarily lack detail, e.g., by taking latitudinal averages, but to me this is fine. The point of this paper seems to be less a thorough analysis of all the potential issues with the SD simulations but rather to provide a helpful starting point for those wishing to analyse output from the SD runs. The paper is well written and I only have minor comments.

Detailed comments:

page 1, line 2: "Here" could be removed

page 10, line 20: "Next" seems a strange way to start sentence here; how about "We therefore also compare ..."

page 10, line 21: I don't quite follow this argument for restricting the average to 60S-60N. If you do the usual cos(lat) area weighting then grid points near the pole are naturally de-emphasised. If there really are issues with a few grid points then restricting it to ~85S-85N would have been more reasonable. The promised discussion about the sensitivity of choice of latitude bounds at the end of this paragraph (section 5) is a bit cursory (in section 5 you basically state that you've looked at the latitudinal distribution and it looks okay – that's probably not the kind of sensitivity analysis that most readers would expect).

page 12: I would find it helpful if you included a bit of discussion on the sources of interannual variability (e.g., due to ENSO, QBO)

page 13, line 22: "not intuitive and" could be removed

page 13, line 25: "worse" -> "greater"

page 13, line 28: "including" -> "included"

page 15, line 6: "better interannual variability" – do you mean "more realistic interannual variability"?

page 17, line 6: "... when inferring dynamics-tracer relationships" – it may be important to clarify that this refers to impact of dynamics on tracers, but not the other way around ("relationship" suggests it could go both ways) ...

page 17, line 23: "including" -> "included"

---

## Author Comment (AC1) · 6 Dec 2019

We have uploaded our response to all three referees' comments, as well as the short comment, in the supplementary file entitled "acp-2019-625-supplement.pdf."

Please also note the supplement to this comment:
https://www.atmos-chem-phys-discuss.net/acp-2019-625/acp-2019-625-AC1-supplement.pdf

---

## Author Response (AR2)

Dr. Clara Orbe
NASA Goddard Institute for Space Studies Code 611 New York · NY 10025
clara.orbe@nasa.gov

Dr. Paul Young
Editor,
Atmospheric Chemistry and Physics

February 1, 2020
re: manuscript number: acp-2019-625 Title: "Description and Evaluation of the Specified-Dynamics Experiment in the Chemistry-Climate Model Initiative (CCMI)"

Dear Dr. Young,

We thank you for your consideration of our responses to the reviews of our manuscript. In addition to the technical edits that you have indicated that we correct (please see the next page), we have also incorporated the feedback from the new reviewer in the most recently updated draft. As in the previous round of revisions we have provided two versions of the revised manuscript, one of which includes the corrections highlighted in red. We hope that the manuscript is now acceptable for publication in ACP. I confirm that my coauthors, David Plummer, Darryn W. Waugh, Huang Yang, Patrick Jockel, Douglas E. Kinnison, Beatrice Josse, Virginie Marecal, Makoto Deushi, Nathan Abraham, Alexander Archibald, Martyn Chipperfied, Sandip Dhomse, Wuhu Feng and Slimane Bekki concur with the submission of our manuscript in its revised form. The revised version of the manuscript has been resubmitted electronically.

Yours sincerely,

Dr. Clara Orbe

**Response to Technical Comments from Editor**

We thank the Editor for providing the list of technical comments which have now been corrected in the most updated version of the manuscript.

Our response to the comments raised are as follows:

1. *"P5, L22: Should the "e.g." be "i.e.", or removed?  In any case, please least all models that this applies to."*

Good point – we have replaced "e.g." with "i.e." as the stated list of models does include all models to which this applies.  Please see the revised text.

2. *"P18, L28: her -> here"*

Thank you for catching this typo.  This has been fixed.

3. *"P20, L15: Should HADLEY be Hadley (as in the Centre)?  If so, please provide a reference as well."*

Our apologies for this confusing reference.  We should have specified that HadISST SSTs and SICs are used.  Please see the revised text (note that HadISST is referenced earlier on in the manuscript so that a new reference is not needed).

4. *"P20, L15: begin -> being"*

Thank you for catching this typo.  This has been fixed.

Response to Reviewer 4

We thank the referee for his/her suggested edits to the manuscript.  Incorporation of this feedback into the new revised manuscript has improved the clarity and readability of the study.

Our response to the comments raised are as follows:

1. *"60S-60N averaging: if your motivation is to focus the analyses on the tropics and mid-latitudes then why not point this out to the reader (I may not be the only one who gets confused about the wording of that sentence, which seems to only refer to problems near the poles – quite a different argument"*

We thank for the reviewer for her/his comment.  We have rephrased this passage by making explicit reference to the fact – as requested by the reviewer – that we are only interested in latitudes equatorward of 60°S and 60°N.   The new passage in the most updated draft now reads as follows:

**"In order to focus our analysis on the tropics and midlatitudes we restrict our analysis of temporal variability to spatial averages performed over latitudes between 60°S and 60°N, with the exception of the vertical velocities v\* and w\*. For the latter variables, which change sign from positive to negative in the subtropics in both the troposphere and stratosphere, we perform averages over 30°S and 30°N. Our exclusion of latitudes outside the range 60°S-60°N is in order to avoid emphasizing the poles, where differences among the simulations may reflect large sensitivities to a few grid points and/or numerical instabilities."**

2.  *"interannual variability: you could remark that while not specifically analyzing for ENSO, QBO, these sources of variability may likely still dominant the signal (true?)"*

We think that this point (i.e. mentioning the explicit contribution of ENSO and QBO to interannual variations in both dynamical and transport quantities) is already clear in the text.  Please see line 16 on page 6:

**"As such, our assessment of interannual variability, which evaluates only the degree of correlation between timeseries, differs from previous studies (Chrysanthou et al. (2019)), in which timeseries were further decomposed in terms of different modes of interannual variability (i.e. the El-Nino Southern Oscillation, the Quasi-Biennial Oscillation, etc.)."**

Therefore, no changes to the text have been made as it is not clear to us why this point needs to be made again in the manuscript.  Doing so would be redundant.

3.  *"MERRA: which one, MERRA-1 or MERRA-2? Please clarify in paper"*

Well, there is no official "MERRA-1" per the notation suggested by the reviewer.  The products are, simply, MERRA and MERRA-2.  Although we feel that we have been clear that we are using

the first version of MERRA via the reference to Rienecker et al. (2011) we have, per the reviewer's request, made this still clearer through the following changes to that section:

"…both constrained with MERRA meteorological fields (Reinecker et al. (2011)) (not MERRA-2 (Gelaro et al. (2017)))."

Note that the new Gelaro et al. (2017) citation has also been included in the references.